# Structural insight into the distinct regulatory mechanism of the HEPN−MNT toxin-antitoxin system in *Legionella pneumophila*

Chenglong Jin [1,2,8], Cha-Hee Jeon[1,3,8], Heung Wan Kim[1], Jin Mo Kang [1,3], Yuri Choi [4], Sung-Min Kang[5], Hyung Ho Lee [4], Do-Hee Kim [6,9] ✉, Byung Woo Han [1,3,9] ✉ & Bong-Jin Lee [2,7,9] ✉

HEPN−MNT, a type VII TA module, comprises the HEPN toxin and the MNT antitoxin, which acts as a nucleotidyltransferase that transfers the NMP moiety to the corresponding HEPN toxin, thereby interfering with its toxicity. Here, we report crystal structures of the *Legionella pneumophila* HEPN−MNT module, including HEPN, AMPylated HEPN, MNT, and the HEPN−MNT complex. Our structural analysis and biochemical assays, suggest that HEPN is a metal-dependent RNase and identify its active site residues. We also elucidate the oligomeric state of HEPN in solution. Interestingly, *L. pneumophila* MNT, which lacks a long C-terminal α4 helix, controls the toxicity of HEPN toxin via a distinct binding mode with HEPN. Finally, we propose a comprehensive regulatory mechanism of the *L. pneumophila* HEPN−MNT module based on structural and functional studies. These results provide insight into the type VII HEPN−MNT TA system.

A toxin-antitoxin (TA) system typically comprises a toxin and a cognate antitoxin commonly encoded by adjacent genes. The first known TA system was discovered in the *Escherichia coli* mini-F plasmid in the 1980s and contributes to the stable maintenance of plasmids[1]; moreover, the daughter cells are killed through the postsegregational killing (PSK) model if they fail to inherit the plasmid copy-encoded TA modules[2]. Subsequently, TA modules encoded on the chromosome were discovered in *E. coli*[3]; these modules are commonly located within bacterial genomic islands or constitute small genomic islets[4]. Over the years, numerous TA systems have been described and classified into eight main categories according to the nature of the antitoxin and the manner in which the antitoxin interacts with the toxin to inhibit its toxicity[5,6]. Among them, the most widely studied type II TA system consists of a protein toxin that acts as an RNase and a protein antitoxin that neutralizes the toxicity of the protein by directly blocking the active site or mediating allosteric regulation of the toxin[7–11].

HEPN (higher eukaryotes and prokaryotes nucleotide-binding) protein and its cognate MNT (minimal nucleotidyltransferase), was initially reported as a type II TA system[12,13]. However, recently, the HEPN−MNT module was reported as a newly identified type VII TA system due to the manner of toxin neutralization in the module[6]. In a previous study, the HEPN−MNT complex was predicted to form a substrate binding site, thereby promoting the nucleotidyltransferase activity of MNT, which can transfer NTP to other substrates, such as proteins, nucleic acids, or small molecules[14]. Subsequent structural studies revealed that HEPN adopts a helix-bundle conformation comprising an $RX_4HXY$ motif as an active site[6] and acts as a metal-independent ribonuclease[13,15]. The cognate MNT adopts an α/β fold that belongs to the Polβ-like NTs, with a $GSX_{10}DXD$ motif, which can

[1]The Research Institute of Pharmaceutical Sciences, College of Pharmacy, Seoul National University, Seoul, Republic of Korea. [2]MasterMediTech, Seoul, Republic of Korea. [3]Natural Products Research Institute, College of Pharmacy, Seoul National University, Seoul, Republic of Korea. [4]Department of Chemistry, College of Natural Sciences, Seoul National University, Seoul, Republic of Korea. [5]College of Pharmacy, Duksung Women's University, Seoul, Republic of Korea. [6]Research Institute of Pharmaceutical Sciences, College of Pharmacy, Sookmyung Women's University, Seoul, Republic of Korea. [7]College of Pharmacy, Ajou University, Suwon, Republic of Korea. [8]These authors contributed equally: Chenglong Jin, Cha-Hee Jeon. [9]These authors jointly supervised this work: Do-Hee Kim, Byung Woo Han, Bong-Jin Lee. ✉e-mail: dohee.kim@sookmyung.ac.kr; bwhan@snu.ac.kr; lbj@nmr.snu.ac.kr

transfer nucleotide monophosphates (NMPs) to its cognate HEPN protein to abolish its toxicity[15,16]. Furthermore, MNT and HEPN from *Shewanella oneidensis* bind with each other at a ratio of 2:6[16], while MNT and HEPN from *Aphanizomenon flosaquae* form a complex at a ratio of 2:2[15]. It was suggested that in these cases, the long C-terminal α4 helix of MNT plays a crucial role in controlling the toxicity of the cognate HEPN toxin. As a result, the α4 helix not only affects the formation of an extensive binding interface but also promotes the AMPylation activity of MNT[15,16].

*Legionella pneumophila*, a main cause of Legionnaires' disease[17], is often fatal to humans if not promptly addressed[18,19]. Until now, only the type II HipBST module, which is related to the host signal transduction pathway[13,20–22], has undergone structural and functional studies among the TA systems of *L. pneumophila*[23]. Moreover, MNT from *L. pneumophila* (MNT[Lpg]) lacks the C-terminal α4 helix due to size limitations (~108 aa for MNT[Lpg], ~139 aa for MntA from *S. oneidensis*[16], and ~150 aa for MNT from *A. flos-aquae*[15]). Thus, it is essential to reveal the molecular mechanism of the HEPN−MNT module in *L. pneumophila*.

Here, we present the structural and functional characterization of HEPN−MNT from *L. pneumophila*. Through structural analysis of dimeric HEPN[Lpg] and biochemical assays, an active site comprising Q44 and E47, which are coordinated by Mg[2+], is discovered. In addition, the metal-dependent HEPN[Lpg] digests the 23S and 16S rRNA from *L. pneumoniae*. Although MNT[Lpg] lacks the C-terminal α4 helix, the poly-AMPylation activity of MNT[Lpg] is verified through in vivo and in vitro enzymatic assays. As a result, MNT[Lpg] specifically transfers three or four AMPs to HEPN[Lpg] using Mg[2+] as a co-factor. Furthermore, the inhibitory mechanisms of HEPN[Lpg] toxicity are identified through analysis of the crystal structures of tetrameric HEPN[Lpg] and AMPylated HEPN[Lpg] at the molecular level. Finally, we reveal the comprehensive regulatory process within the HEPN[Lpg]-MNT[Lpg] module, which could provide insight into the type VII TA system.

## Results

### Overall structure of HEPN from *L. pneumophila*

To investigate the function of HEPN[Lpg], our initial approach was to obtain the tertiary structure of HEPN[Lpg] WT (henceforth, tetrameric HEPN[Lpg]): the crystal structure was determined at a resolution of 1.8 Å. Four molecules were found in one asymmetric unit (ASU), which consists of two homodimers (chains A/B and chains C/D) (Fig. 1a). Specifically, five interfaces exist in this homotetramer form: two dimer interfaces and three tetramer interfaces (chains A/B and chains C/D for dimer interfaces; chains B/C; chains A/C; and chains A/D for tetramer interfaces). A comparison of the buried surface area as a percentage of the entire surface area indicated that the dimeric interface is generally larger than the tetrameric interface (Supplementary Table 1). Specifically, the dimerization interface between chain A and chain B is mainly composed of the α1 and α2 helices and the α4−α5 loop (henceforth, Y-loop) (Fig. 1a, b). Among them, the residues in the cross-shaped α2 helices contributed to the majority of interaction networks of dimer interfaces. Specifically, residue Q44 in the α2 helix and residues R17 and Q21 in the α1 helix interact with T101, T104, Y105, and N106 in the Y-loop via hydrogen bonds. Furthermore, residue L37 in the α2 helix participates in hydrophobic interactions with several residues, namely, A25, L28, L34, and L42 (Fig. 1b).

Moreover, additional interfaces between two homodimers (chains A/B and chains C/D) exist in the tetrameric conformation; these two interfaces consist of α2 and α3 helices and α2−α3 loops (Fig. 1c). Notably, Q64 and R73 interact with neighboring residues via strong and extensive hydrophilic interactions. Therefore, residues Q64 and R73 are suggested to have crucial functions in tetrameric assembly. To verify the oligomeric state of HEPN[Lpg] in solution, analytical ultra-centrifugation (AUC) experiments were conducted using HEPN[Lpg] WT and its mutants (Q64A and R73A). The results showed that HEPN[Lpg] WT existed in both the dimeric and tetrameric states, whereas both

HEPN[Lpg] mutants predominantly existed in dimeric states (Fig. 2a−c). Additionally, to quantify the homotetramerization of HEPN[Lpg], surface plasmon resonance (SPR) experiments were performed. Interestingly, the kinetic sensorgrams indicated that the tetramerization between ligands and analytes exhibited considerable fast association and dissociation rates (Fig. 2d−f), suggesting that the self-assembly process is transient and reversible. However, due to the rapid kinetics, kinetic affinity analysis was not appropriate. Instead, the maximal response units ($R_{max}$) were analyzed using a steady-state affinity model (Fig. 2g−i). As the molecular weights (MWs) of HEPN[Lpg] WT and its mutants (Q64A and R73A) are nearly identical, the $R_{max}$ values were proportional to the binding affinity between the ligands and analytes. As a result, the $R_{max}$ value for HEPN[Lpg] WT was significantly higher than that of the HEPN[Lpg] mutants (Fig. 2g−i), indicating that HEPN[Lpg] WT has a greater tendency for self-assembly compared to the mutants. Taken together, these results indicate that HEPN[Lpg] exists in a balanced equilibrium between two oligomeric states in solution and that Q64 and R73 act as key residues in switching the oligomeric state of HEPN[Lpg].

### HEPN[Lpg] possesses a distinct catalytic site

Next, to identify the conserved motif of HEPN[Lpg], sequence alignment of HEPN[Lpg] with several structural homologs was conducted. The results revealed that HEPN[Lpg] and its homologs share a highly conserved RX$_4$HXY motif (where X is any amino acid), even though the sequence identity is relatively low (Fig. 3a). In a previous study, the HEPN superfamily, which comprises only α-helical domains, was shown to act as a metal-independent ribonuclease by using the RX$_4$HXY motif, and the tyrosine within this motif served as an acceptor site for NMP from its cognate MNT[15,16]. Furthermore, the RNA digestion activity of HEPN was studied. Therefore, we focused on the RNA cleavage activity of HEPN[Lpg] against total RNA (rRNA and tRNA) extracted from *L. pneumophila*. The results showed that purified HEPN[Lpg] was capable of cleaving 23S rRNA and 16S rRNA into small fragments (Fig. 3b and Supplementary Fig. 1a). Additionally, HEPN[Lpg] could also digest the corresponding 23S rRNA and 16S rRNA from the intact 70S ribosome. In this case, some of the digested RNA remained as relatively large fragments (Supplementary Fig. 1b). This result might be due to the rRNA being tightly packed and protected by ribosomal proteins, rendering many regions inaccessible to RNases.

As a next step, we tried to identify the active site of HEPN[Lpg]. Given the RX$_4$HXY motif is located within the Y-loop, its conformation might be influenced by homotetramerization of HEPN[Lpg]. Therefore, we determined the crystal structure of HEPN[Lpg] Q64A (henceforth, dimeric HEPN[Lpg]). As a result, the triangular homodimer HEPN[Lpg] is formed by a helical bundle of two monomers, in which a deep cleft is formed by three alpha helices (α2, α3 and α4) from each subunit (Fig. 3c). Moreover, the RX$_4$HXY motif is situated within the Y-loop, which is positioned at the base of the cleft. Among the residues in the RX$_4$HXY motif, the first arginine residue and the sixth histidine residue are suggested to cooperate with each other in metal-independent RNase activities[15]. However, in dimeric HEPN[Lpg], the corresponding residues R98 and H103 from each subunit are located far from each other (the distance between Nε2 of H103 and NH2 of R98 is ~10.9−11.0 Å, and the distance between Nε2 of H103 in separate subunits is ~23.1 Å) (Fig. 3c). Notably, the overall structures of the HEPN[Lpg] dimers from the crystal structures of tetrameric HEPN[Lpg] and dimeric HEPN[Lpg] are highly similar, with a root mean square deviation (RMSD) of 0.383 Å for 226 Cα atom pairs. Specifically, the Y-loop region does not undergo significant structural changes between the two oligomeric states, with residues R98 and H103 aligning closely with each other (Supplementary Fig. 2a). Moreover, residue root mean square fluctuation (RMSF) was calculated via molecular dynamics (MD) simulation. The analysis revealed that residues in the Y-loop region exhibited RMSF values ranging from 0.5 to 0.7 Å (Supplementary Fig. 2b), indicating that the Y-loop in

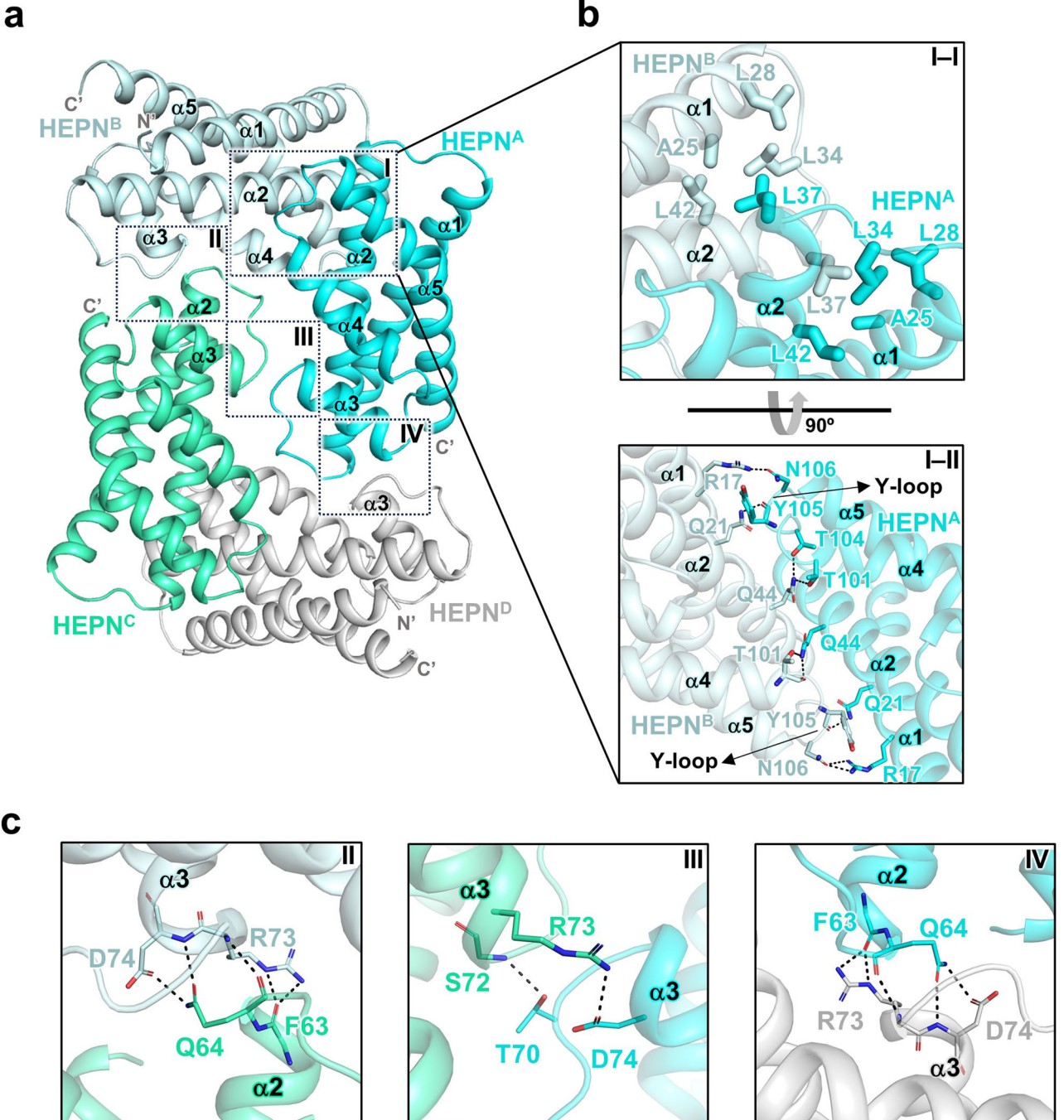

**Fig. 1 | Overall structure of HEPN$^{Lpg}$. a** Overall structure of the tetrameric HEPN$^{Lpg}$. Chains A, B, C and D are shown in cyan, pale cyan, green cyan, and gray, respectively. The regions involved in dimer and tetramer interfaces are marked with squares and labeled. **b** The regions containing hydrophobic interactions and hydrogen bonds at the dimer interface are depicted in I–I and I–II, respectively. The residues involved in the dimer interface between chains C and D are not represented here due to their identity with those between chains A and B. **c** The residues involved in hydrogen bonds in tetramer interfaces are shown in II, III, and IV.

HEPN$^{Lpg}$ is relatively rigid and less dynamic. As previously mentioned, the residues T101, T104, Y105, and N106 participate in hydrophilic interactions with several residues from the counterpart chain, which contributes to the reduced flexibility of the Y-loop (Fig. 1b). Overall, the RX$_4$HXY motif is unlikely to serve as the catalytic site in HEPN$^{Lpg}$ due to specific structural characteristics of the highly conserved residues.

Intriguingly, a putative metal-coordinated site was observed in the crystal structure of dimeric HEPN$^{Lpg}$ (Fig. 3d). Specifically, a metal ion was found to be coordinated by two Q44 residues, two E47 residues and two water molecules, which constitute a solvent-accessible surface area located in the midst of a dimerization-driven cleft. Thus, it was

rational to hypothesize that the metal coordination site may exert a specific influence on the enzymatic activity of HEPN$^{Lpg}$. To confirm this, in vitro RNase assays and spot-plating assays were conducted. First, ribonuclease activity was measured for dimeric HEPN$^{Lpg}$ and its mutants (for highly conserved residues in the RX$_4$HXY motif, R98A and H103A; for the metal coordination site, Q44A, E47A, and Q44AE47A); additionally, the dimeric HEPN$^{Lpg}$ was treated with 10 mM EDTA to remove the metal ions. As a result, Q44A, E47A, and Q44AE47A showed significant decreases in RNase activity compared to that of dimeric HEPN$^{Lpg}$, while R98A and H103A had negligible impacts on RNase activity (Fig. 4a). Moreover, comparing the RNase activities between

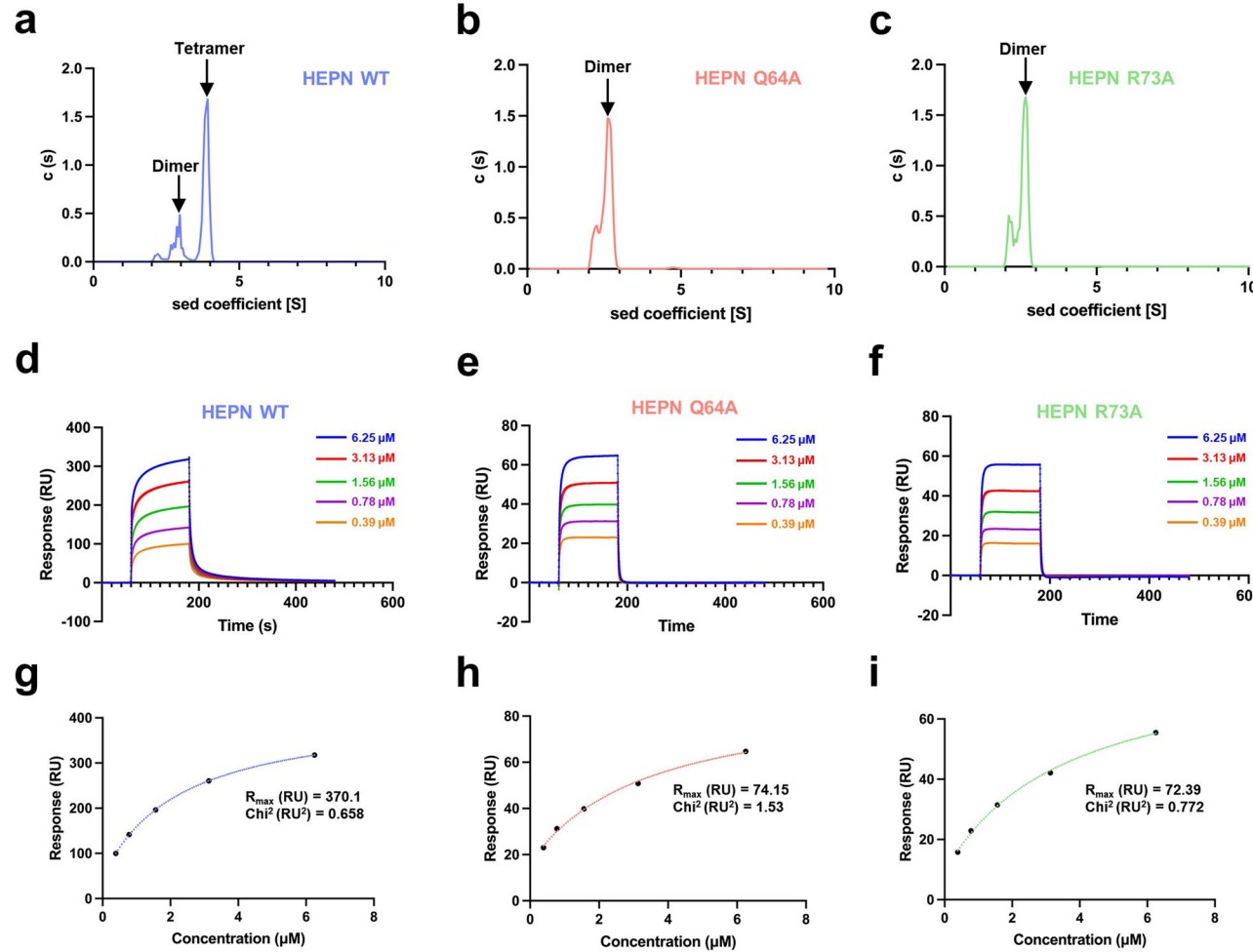

**Fig. 2 | Homotetramerization capability of HEPN$^{Lpg}$. a–c** Analytical ultracentrifugation (AUC) experiments. The plots show the sedimentation coefficient distributions $c(s)$ of various HEPN$^{Lpg}$ samples (HEPN$^{Lpg}$ WT, Q64A, and R73A) analyzed by AUC. HEPN$^{Lpg}$ WT exists in both dimeric (~3S) and tetrameric (~4S) states, whereas HEPN$^{Lpg}$ Q64A and R73A exist only in the dimeric state (~3S). Arrows indicate the peak positions of the dimeric and tetrameric species. **d–i** Surface plasmon resonance (SPR) experiments. Kinetic sensorgrams for homotetramerization of HEPN$^{Lpg}$ WT (**d**), HEPN$^{Lpg}$ Q64A (**e**), and HEPN$^{Lpg}$ R73A (**f**). Each colored line represents the binding response at the respective concentration, demonstrating the rapid association and dissociation phases for HEPN$^{Lpg}$ WT (**d**), HEPN$^{Lpg}$ Q64A (**e**), and HEPN$^{Lpg}$ R73A (**f**). Saturation binding curves for homotetramerization of HEPN$^{Lpg}$ WT (**g**), HEPN$^{Lpg}$ Q64A (**h**), and HEPN$^{Lpg}$ R73A (**i**). The response units (RU) plotted against different concentrations (μM) of analytes are derived from the kinetic sensorgrams, which are further used to determine the maximum RU ($R_{max}$) using a nonlinear regression curve.

Ribolock-treated and untreated samples showed almost no variation in RFU, indicating that the observed RNase activity in the HEPN$^{Lpg}$ samples is unlikely to result from contamination by other RNases (Supplementary Fig. 3). Furthermore, the removal of metal ions almost completely abolished the RNase activity of dimeric HEPN$^{Lpg}$, and Mg$^{2+}$ was the sole metal ion that rescued the RNase activity of the EDTA-treated dimeric HEPN$^{Lpg}$ (Fig. 4b). To further determine the catalytic properties of HEPN$^{Lpg}$, $k_{cat}/K_m$ values were measured using Michaelis–Menten kinetics (Supplementary Fig. 4 and Supplementary Table 2). The results showed that, compared with that of dimeric HEPN$^{Lpg}$, EDTA-treated HEPN$^{Lpg}$ and HEPN$^{Lpg}$ mutants (Q44A, E47A, and Q44AE47A) exhibited reductions in $k_{cat}/K_m$. Moreover, dilution spot assays confirmed that the induction of dimeric HEPN$^{Lpg}$, R98A, and H103A in *E. coli* cells significantly arrested cell growth on plates. In contrast, cell growth was rescued when Q44A, E47A, and Q44AE47A were expressed in vivo (Supplementary Fig. 5). Taken together, these results indicated that the Mg$^{2+}$-coordinated site is indeed the active site of HEPN$^{Lpg}$ ribonuclease. Furthermore, it is worth noting here that the α2–α3 loop, which inserts into the deep cleft of the HEPN$^{Lpg}$ dimer, precisely crosses the cleft comprising the active site (Fig. 3e) in tetrameric HEPN$^{Lpg}$. This effect might trigger steric hindrance between the

RNA substrates and the HEPN$^{Lpg}$ active site, resulting in the inhibition of HEPN$^{Lpg}$ activity.

## Compared with its structural homologs, MNT$^{Lpg}$ lacks an α4-helix

To determine the function of the cognate MNT$^{Lpg}$, the crystal structure was determined at a resolution of 2.79 Å. Although the refined model accounts for two monomers that form a homodimer in ASU, the SEC-MALS result indicated that MNT$^{Lpg}$ was monomeric in solution (Fig. 5a and Supplementary Fig. 6). It is suggested that the dimeric structure was formed by crystal packing, and MNT$^{Lpg}$ is anticipated to function in its monomeric state. MNT$^{Lpg}$ adopts a polβ-like fold consisting of a four-stranded β-sheet flanked by two α-helices on one face of the sheet and one α-helix on the opposite face (Fig. 5b). Notably, a long glycine-rich loop (G-loop) exists between the antiparallel β1 and β2 strands and is directed toward the solvent-exposed region of MNT$^{Lpg}$ (Fig. 5b, c). Structural comparison was performed using the DALI server[24], which revealed that several structural homologs, namely, MNT from *S. oneidensis* (PDB ID 5YEP chain A, Z score = 9.5), MNT from *A. flos-aquae* (PDB ID 7AE2 chain B, Z score = 8.7), and MNT from *Haemophilus influenzae* (PDB ID 1NO5 chain A, Z score = 15.1), exhibited high

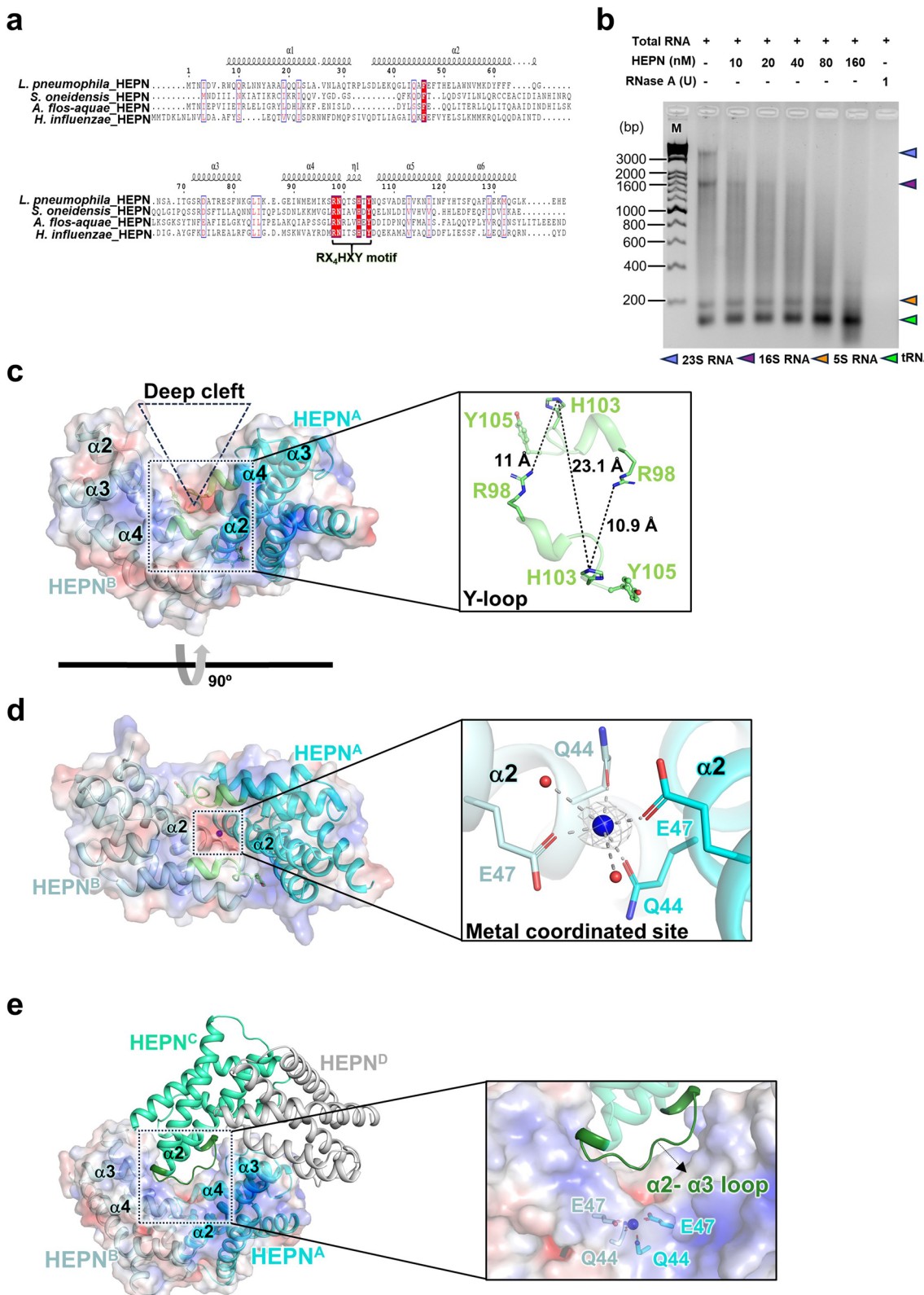

structural similarity. Sequence alignment of the structural homologs revealed that MNT^Lpg possesses a highly conserved motif, which is named the GSX10DXD (where X is any amino acid) motif (Fig. 5d). Intriguingly, the GSX10DXD motif is located precisely near the G-loop (Fig. 5b, c). In a previous study, polβ-like enzymes with the GSX10DXD motif were shown to contribute to the activity of nucleotidyl-transferases, which are involved in modifying the corresponding HEPN

in bacterial cells[15,16]; in this case, the C-terminal α4 helix of MNT indeed played a role in HEPN inhibition. For example, the α4 helix could directly insert into the deep cleft formed by the HEPN dimer, resulting in blockage of the active site of HEPN[13]. Additionally, the α4 helix was involved in docking to the α1 and α6 helices of HEPN to form a large binding interface, and MNT tended to lose its AMPylation ability when the α4 helix of MNT was removed[15]. Overall, the lack of the α4 helix

**Fig. 3 | Active site of HEPN$^{Lpg}$. a** Sequence alignment of HEPN$^{Lpg}$ with its homologs. The secondary structure of HEPN$^{Lpg}$ is depicted in the upper region. The conserved and similar residues are represented as red and white boxes, respectively. The highly conserved RX$_4$HXY motif is highlighted and labeled. **b** In vitro total RNA digestion assay. HEPN$^{Lpg}$ cleaves the 23S rRNA and 16S rRNA from *L. pneumophila*. HEPN$^{Lpg}$ Q64A (henceforth, dimeric HEPN$^{Lpg}$) is used for this assay, with RNase A serving as a positive control. **c** The side view of the dimeric HEPN$^{Lpg}$ is shown in surface and cartoon representations. The highly conserved RX$_4$HXY motif located at the Y-loop is highlighted in lime, and the deep cleft region is indicated by a dashed triangle. Close-up view of the Y-loop with the presumed active site residues

(R98 and H103) shown in stick representation. Y105 is highlighted with a sphere. The distances between conserved residues were measured using *PyMOL*. **d** The top view of the dimeric HEPN$^{Lpg}$ is shown to the same as that in Fig. 3c. Close-up view of the metal-coordinated site; Mg$^{2+}$ and water molecules are shown as blue and red spheres, respectively, whereas the hexahedral coordination of Mg$^{2+}$ is represented by gray dotted lines. The coordinated residues are shown in stick representation. **e** Tetrameric HEPN$^{Lpg}$ is depicted in surface and cartoon representations; the HEPN$^{Lpg}$ dimer comprising chains A and B is shown identically to that in Fig. 3c. The α2–α3 loop is highlighted in dark green. Close-up view showing that the α2–α3 loop causes steric hindrance at the active site of HEPN$^{Lpg}$.

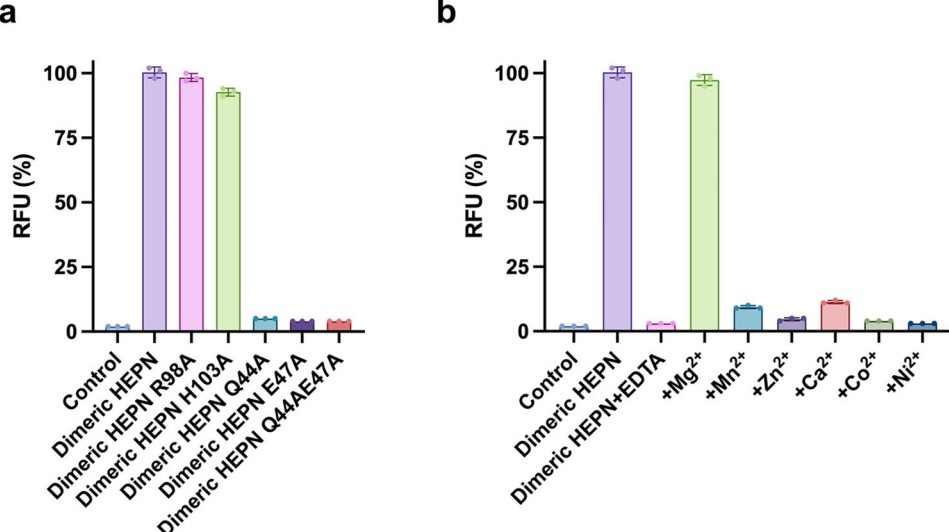

**Fig. 4 | In vitro RNase activity assay. a** RNase activity assay of dimeric HEPN$^{Lpg}$ and its mutants. **b** Effect of the presence of metal ions on RNase activity. Only Mg$^{2+}$ supports the RNase activity of HEPN$^{Lpg}$. RNase activity is represented as the proportion of each sample to that of dimeric HEPN$^{Lpg}$, where dimeric HEPN$^{Lpg}$ is taken as 100%. The data are presented as the average of three independent replicates with standard deviation (±SD), and each corresponding data point is represented as a dot plot.

could impair complex formation and modification reactions. Notably, MNT$^{Lpg}$ lacks the α4 helix at the C-terminus (Fig. 5b); in this regard, it is rational to examine whether MNT$^{Lpg}$ might inhibit the HEPN$^{Lpg}$ toxin through nucleotidyltransferase or other mechanisms.

### HEPN$^{Lpg}$ could be poly-AMPylated by MNT$^{Lpg}$ in vivo and in vitro

To further investigate the ability of MNT$^{Lpg}$, which lacks the α4 helix, to function as a nucleotidyltransferase, several experiments were carried out. First, an ~1 kDa band shift of HEPN$^{Lpg}$ coexpressed with MNT$^{Lpg}$ in *E. coli* was observed via SDS–PAGE, while the sole band appeared in cells expressing HEPN$^{Lpg}$ alone or coexpressing HEPN$^{Lpg}$ Y105A (in the Y-loop region) with MNT$^{Lpg}$ in identical bacterial strains (henceforth, the increased protein is termed NMPylated HEPN$^{Lpg}$, and the unmodified protein is termed apo HEPN$^{Lpg}$) (Fig. 6a). Additionally, the accurate MWs of the purified apo and NMPylated HEPN$^{Lpg}$ proteins were measured using intact-MS. The results revealed that the MW of apo HEPN$^{Lpg}$ was 16693.2 Da, which is equal to the theoretical MW of HEPN$^{Lpg}$, while the MW of NMPylated HEPN$^{Lpg}$ was increased by 971.4–1277.2 Da compared to that of apo HEPN$^{Lpg}$ (Fig. 6b). The increase in MW was speculated to correspond to the MW of three or four NMPs; however, the exact NMP type could not be verified. Then, phosphodiesterase 1 (PDE1) was used to cleave the phosphodiester bonds between the hydroxyl group and phosphate group of the NMPs. Afterward, the small molecules were collected and analyzed by LC–MS. The results showed that the MW of the small molecules was 348.22 Da, which corresponds to the MW of AMP (Supplementary Fig. 7).

To further investigate the poly-AMPylation ability of MNT$^{Lpg}$, in vitro NMPylation assays were performed. First, HEPN$^{Lpg}$ was subjected to reactions with MNT$^{Lpg}$ and ATP substrates. As a result, poly-AMPylation of HEPN$^{Lpg}$ was exclusively observed upon addition of

HEPN$^{Lpg}$ WT to the reactions (Fig. 6c and Supplementary Fig. 8a). Notably, the production yield of modified HEPN$^{Lpg}$ remained relatively low, despite the inclusion of sufficient ATP substrate in the reaction. The results showed that the AMPylation efficacy of MNT$^{Lpg}$ was insufficient to achieve full AMPylation of the cognate HEPN$^{Lpg}$, which is consistent with the results of the in vivo AMPylation assay. In addition, the nucleotide specificity of MNT$^{Lpg}$ was confirmed using other nucleotide triphosphates (CTP and GTP). The results showed that a negligible band shift occurred when CTP was used as the substrate (Supplementary Fig. 8b), while no modification of HEPN$^{Lpg}$ occurred in the presence of GTP (Supplementary Fig. 8c). Thus, MNT$^{Lpg}$ specifically AMPylates HEPN$^{Lpg}$ using ATP as a substrate. To confirm the active site residues of MNT$^{Lpg}$, several mutants at the highly conserved motif (GSX$_{10}$DXD) were constructed, namely, G36AS37T, D48E, D50E, and D48ED50E. The results showed that there was no modification of HEPN$^{Lpg}$ in any of the mutants (Fig. 6d), which indicated that these residues are definitely involved in the AMPylation reactions. In addition, according to a previous study, polβ-like enzymes can act as nucleotidyltransferases using divalent metal ions as cofactors[25]. To verify this ability, several divalent metal ions were added to EDTA-treated MNT$^{Lpg}$. The results revealed that MNT$^{Lpg}$ functions as a nucleotidyltransferase when Mg$^{2+}$ is added (Fig. 6e). Collectively, these findings demonstrated that MNT$^{Lpg}$ could indeed poly-AMPylate HEPN$^{Lpg}$ by using Mg$^{2+}$ as a co-factor.

### Structural snapshot of AMPylated HEPN$^{Lpg}$

Next, to elucidate the molecular details of NMPylated HEPN$^{Lpg}$, a high-resolution crystal structure was obtained at a resolution of 1.65 Å. The refined model showed that NMPylated HEPN$^{Lpg}$ has a homotetramer structure assembled from two homodimers, which is consistent with

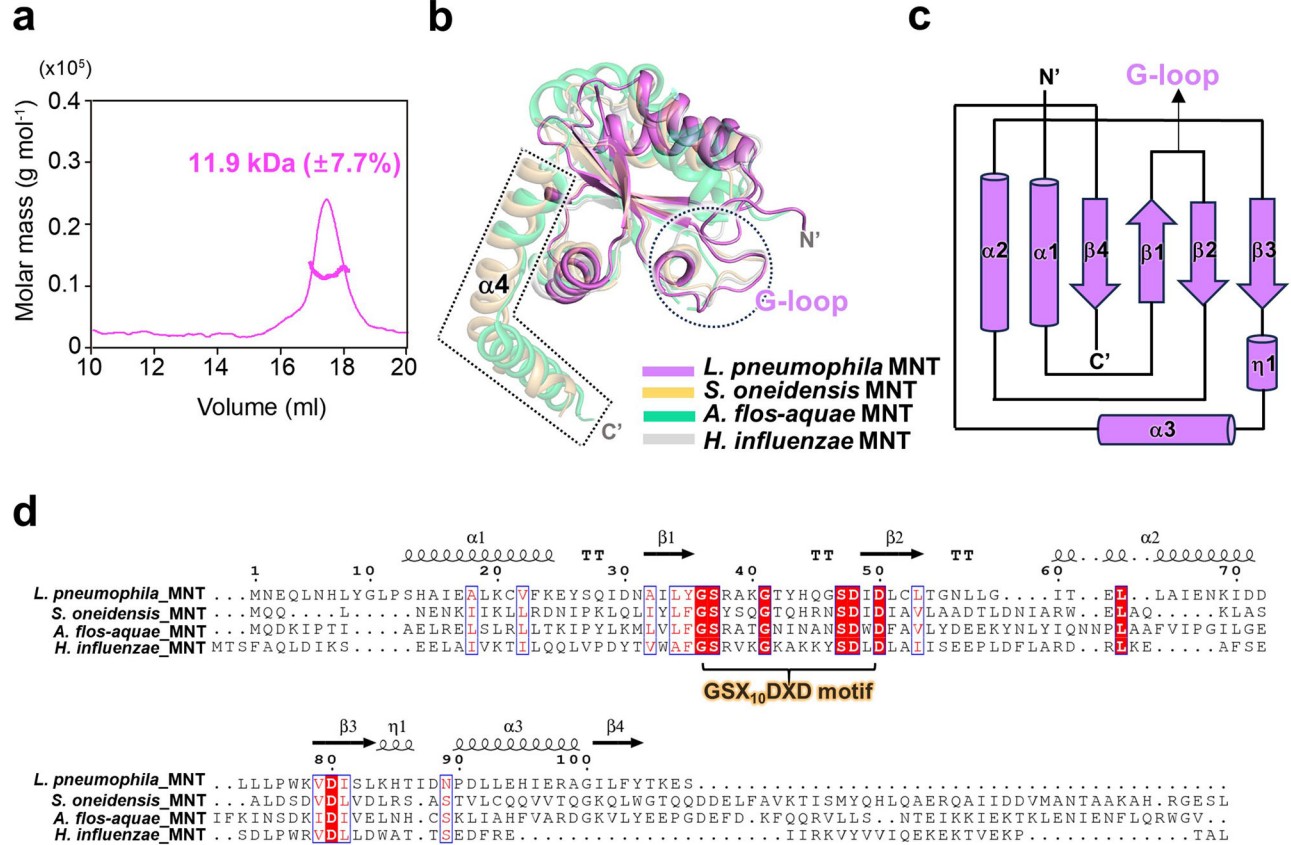

**Fig. 5 | Structural characteristics of MNT$^{Lpg}$. a** The SEC-MALS result shows that MNT$^{Lpg}$ exists as a monomer in solution. **b** Overlay of the MNT$^{Lpg}$ structure with those of its homologs. *L. pneumophila* MNT (violet), *S. oneidensis* MNT (light orange), *A. flos-aquae* MNT (lime green) and *H. influenzae* MNT (gray) are shown in cartoon representation. The G-loop region is indicated by a dashed circle, while the α4 helices of MNT from *S. oneidensis* and *A. flos-aquae* are indicated by dashed boxes. **c** Diagram of the secondary structure of MNT$^{Lpg}$. The G-loop region is located between the β1 and β2 strands. **d** Sequence alignment of MNT$^{Lpg}$ with its homologs. The secondary structure and conserved residues are presented in the same manner as in Fig. 3a. A GSX$_{10}$DXD motif is highly conserved and located in the G-loop region.

the crystal structure of tetrameric HEPN$^{Lpg}$ mentioned above (Fig. 7a). Notably, all of the Y105 residues were modified with AMP through covalent bonds. However, only one AMP could be observed in this crystal structure because of poor electron density maps, which might be caused by the flexibility of the conformation of the AMP attached to Y105. Specifically, covalently linked AMP interacts with N106 and H103 from HEPN$^{Lpg}$ through hydrogen bonds and π-π stacking, and R17 and Q21 still interact with the residues from the corresponding Y-loop site to stabilize it (Fig. 7b). Moreover, several positively charged residues, such as R7, R11 and R17, are positioned at the base of the deep cleft and contribute to the formation of a positively charged crevice, where RNA molecules may bind through charge–charge interactions[26]. However, the negatively charged phosphate of AMP is located near the positively charged crevice, resulting in steric hindrance at the RNA binding site and inhibition of HEPN$^{Lpg}$ activity.

### Crystal structural insight into the MNT$^{Lpg}$-HEPN$^{Lpg}$ complex

Then, to elucidate the mechanistic action of the MNT$^{Lpg}$-HEPN$^{Lpg}$ TA system based on its structure, we determined the crystal structure of the MNT$^{Lpg}$-HEPN$^{Lpg}$ complex at a resolution of 2.4 Å. The results revealed that one MNT$^{Lpg}$ monomer interacts with one HEPN$^{Lpg}$ tetramer in ASU (Fig. 8a), and the HEPN$^{Lpg}$ tetramer from the MNT$^{Lpg}$-HEPN$^{Lpg}$ complex closely resembles that observed in the absence of MNT$^{Lpg}$ (with an RMSD of 0.99 Å for 480 Cα atom pairs). In the complex structure, MNT$^{Lpg}$ interacts with HEPN$^{Lpg}$ via two interaction surfaces (Fig. 8a and Supplementary Table 1). Specifically, one interface is formed by the α2 helix of MNT$^{Lpg}$ and the α3 and α4 helices of HEPN$^{Lpg}$ (henceforth, Interface I), and the other interface is formed by the

α2–β3 loop of MNT$^{Lpg}$ and the α3 helix of HEPN$^{Lpg}$ (Interface II). Several residues at these interfaces are involved in the formation of multiple hydrogen bonds and salt bridges (Fig. 8c). Among these residues, R73 of HEPN$^{Lpg}$ is the key residue that contributes to extensive and strong hydrophilic interactions with MNT$^{Lpg}$. However, the MNT$^{Lpg}$-bound residues of HEPN$^{Lpg}$ in Interface I and II are positioned near each other (the distance could be ~9.2 Å on the basis of the Cα atom of R73), while at the opposite site, the corresponding residues of HEPN$^{Lpg}$ from chains B and chain D are distant from each other (the distance is ~42.2 Å on the basis of the Cα atom of R73). (Supplementary Fig. 9a, b). In this regard, MNT$^{Lpg}$ cannot simultaneously interact with the residues from chain B and chain D of HEPN$^{Lpg}$ in the same manner as in chain A and chain C. Furthermore, native-PAGE and size-exclusion chromatography (SEC) experiments demonstrated that MNT$^{Lpg}$ binds to HEPN$^{Lpg}$ exclusively in its tetrameric state, rather than dimeric state, in solution (Fig. 8d and Supplementary Fig. 9c), consistent with the crystal structure. Taken together, these findings suggested that MNT$^{Lpg}$ could bind solely to tetrameric HEPN$^{Lpg}$ at the adjacent interface sites at a stoichiometric ratio of 1:4.

## Discussion

### HEPN$^{Lpg}$ acts as a Mg$^{2+}$-dependent ribonuclease

A structural study of HEPN$^{Lpg}$ revealed that HEPN$^{Lpg}$ folds into a five-helix bundle containing a highly conserved RX$_4$HXY motif. In a previous study, several HEPN family proteins were shown to function as metal-independent RNases[27]; and the recently discovered HEPN toxin from type VII TA systems has been implicated in digesting tRNA and *ompA* mRNA[12] by using RX$_4$HXY motif. HEPN$^{Lpg}$ efficiently cleaved both

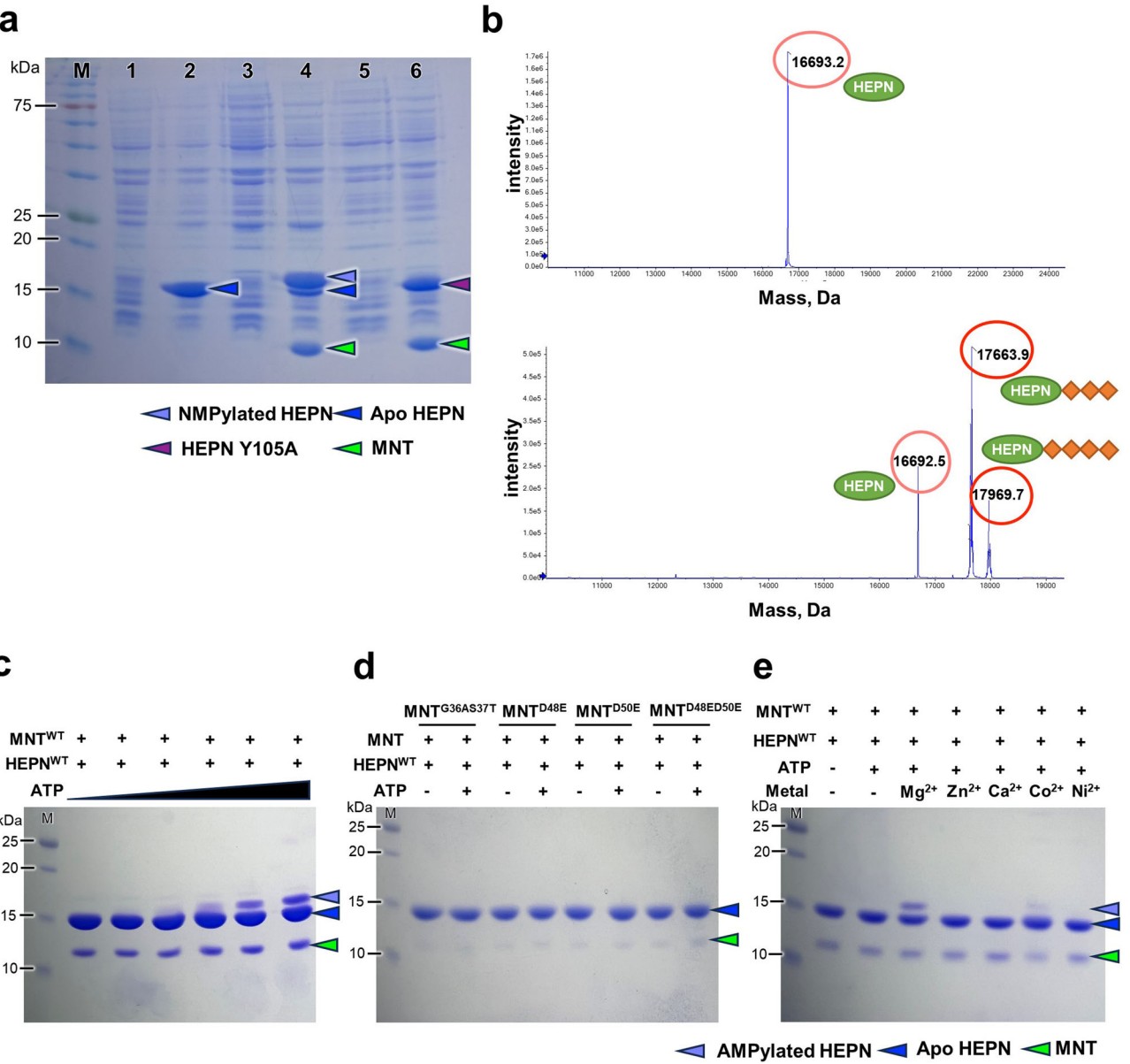

**Fig. 6 | Nucleotidyltransferase activity of MNT^Lpg. a** Expression test using the pET-28a and pET-21a vectors in *E. coli* BL21 (DE3)-codon plus cells. pET-28a-HEPN^Lpg expressed alone (lane 2). pET-28a-HEPN^Lpg and pET-21a-MNT^Lpg coexpressed (lane 4). pET-28a-HEPN^Lpg (Y105A) and pET-21a-MNT^Lpg coexpressed (lane 6). The corresponding preinduced samples are shown in lanes 1, 3 and 5, respectively. **b** The MWs of apo HEPN^Lpg and NMPylated HEPN^Lpg were analyzed via intact-MS. Apo HEPN^Lpg and NMPylated HEPN^Lpg were further purified using the sample from Fig. 6a lanes 2 and 4, respectively. A green circle indicates HEPN^Lpg, whereas an orange square indicates NMP. **c–e** In vitro AMPylation assay. HEPN^Lpg WT undergoes AMPylation by MNT^Lpg WT in an ATP concentration-dependent manner (**c**). Mutations in the G-loop region of MNT^Lpg eliminate the AMPylation ability of MNT^Lpg (**d**). EDTA-treated MNT^Lpg WT exhibited AMPylation ability when Mg²⁺ was used as a co-factor (**e**).

isolated and intact ribosomal 23S rRNA and 16S rRNA in a non-specific manner, thereby disrupting bacterial translational processes. However, in this study, we were unable to determine whether HEPN^Lpg can cleave any specific mRNA. Thus, it is difficult to rule out the possibility that HEPN^Lpg cleaves mRNA in a non-specific cleavage pattern; this investigation will require further research.

Intriguingly, the residues R98 and H103 of RX₄HXY motif in HEPN^Lpg cannot cooperate with each other to constitute an active site. In this regard, the polar residues within this conserved motif might contribute to interactions with the ATP substrates during the nucleotidyl transfer process. In this study, a metal-coordinated site consisting of residues Q44 and E47 and Mg²⁺ was suggested to be the active site of HEPN^Lpg on the basis of biochemical assays. In this regard, when an RNA molecule arrives adjacent to the HEPN^Lpg active site, a water molecule coordinated by Mg²⁺ can be activated, which

subsequently attacks the phosphodiester bond in the RNA molecule. The coordinated Mg²⁺ plays a crucial role in facilitating this hydrolysis reaction by polarizing the water molecule and stabilizing the transition state of the reaction[28]. There are two classes of HEPN−MNT modules in bacteria and archaea, namely, class I and class II[16]. Among them, only HEPN from class I HEPN−MNT modules possesses a canonical RX₄HXY motif necessary for functioning as a ribonuclease. Together, our data suggest that HEPN^Lpg is a metal-dependent ribonuclease.

## Molecular basis of oligomeric state transition in HEPN^Lpg

HEPN^Lpg exists in equilibrium between two oligomeric states in solution, namely, the dimeric and tetrameric states. A comparison of HEPN^Lpg with its homologs suggested that they share a common structural feature, with the α2 helix serving as a primary locus for

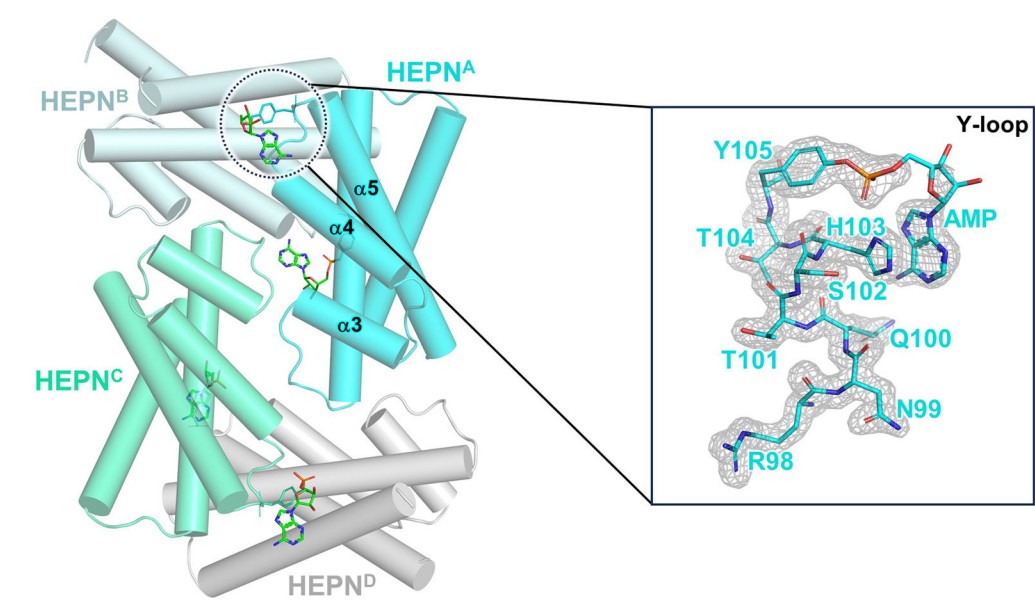

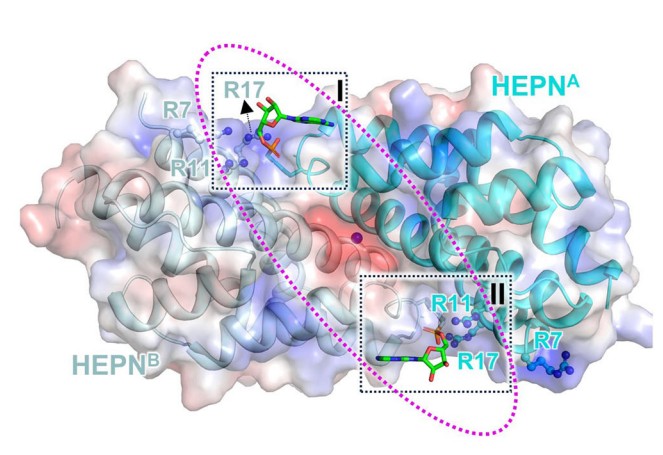

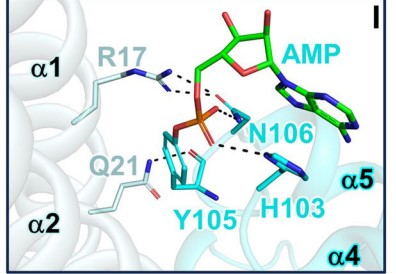

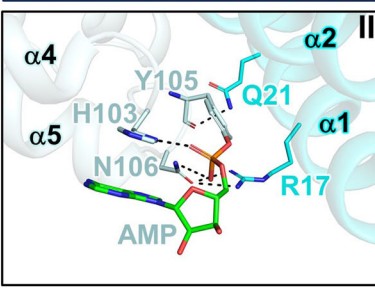

**Fig. 7 | Overall structure of AMPylated HEPN^Lpg. a** Crystal structure of AMPylated HEPN^Lpg. Each of the chains is depicted as straight cylindrical helices with colors corresponding to those in Fig. 1a. The AMP molecules are represented as sticks. AMPylation occurred on the Y105 residue across all four chains. Close-up view of the modified region with an electron density map. The 2Fo-Fc omit map is contoured at the level of 1σ. The hydroxyl group in Y105 forms a covalent linkage with the AMP moiety. **b** The AMPylated HEPN^Lpg dimer is depicted in the same manner as in Fig. 3d. The AMP molecules are shown in stick representation. The modified regions from chains A and B are marked as squares with each label, while the potential RNA binding site is illustrated as a purple dashed circle. The regions containing several interactions are depicted in close-up views of I and II. The residues and AMPs are shown in stick representation. The hydrogen bonds in these regions are represented by dashed black lines.

dimerization (Supplementary Fig. 10a–c). In this regard, no crystal contacts with symmetric mates were observed in the crystal structures of tetrameric HEPN^Lpg and dimeric HEPN^Lpg. Moreover, the angles between the α2 helices were nearly identical in HEPN^Lpg in both states. This result suggests that the tetramerization does not significantly influence the angle between the α2 helices. Notably, the angle between the α2 helices affects the volume of the deep cleft formed by the HEPN dimer. Interestingly, the angles between the α2 helices differ. For example, the angle in HEPN^Lpg is 85.26°, whereas those in *S. oneidensis* HEPN and *A. flos-aquae* HEPN are 72.38° and 76.37°, respectively. In

HEPN^Lpg, the wider angle between the α2 helices could provide enough volume for the α2–α3 loop to insert into the deep cleft (Supplementary Fig. 10d). Nevertheless, a narrow angle of approximately 9–13° in HEPN from *S. oneidensis* and *A. flos-aquae* reduces the volume of the cleft, possibly constraining the space for the α2–α3 loop to insert. Furthermore, when we modeled the tetrameric state of HEPN from *S. oneidensis* and *A. flos-aquae* based on tetrameric HEPN^Lpg, a steric clash occurred adjacent to the α2–α3 loop (Supplementary Fig. 10e, f). Taken together, these findings might explain why these HEPN homologs exist exclusively as homodimers.

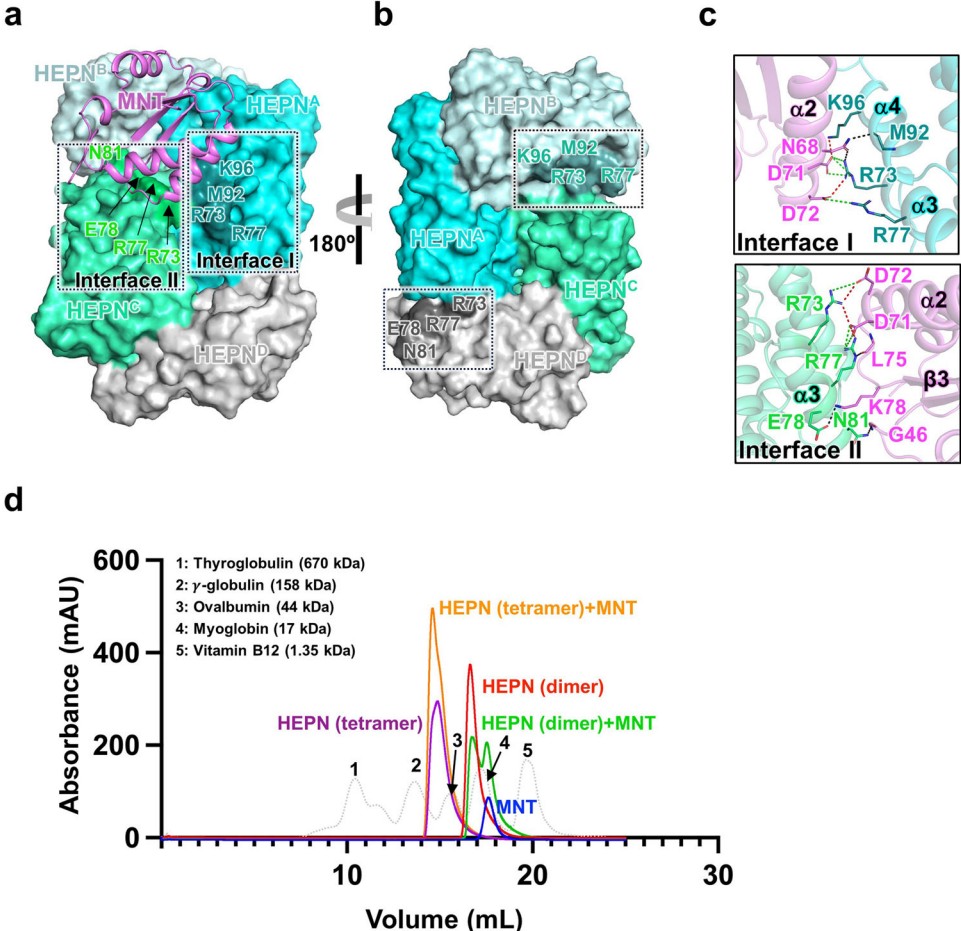

**Fig. 8 | Overall structure of the HEPN$^{Lpg}$-MNT$^{Lpg}$ complex. a**, **b** Crystal structure of the HEPN$^{Lpg}$-MNT$^{Lpg}$ complex. MNT$^{Lpg}$ and HEPN$^{Lpg}$ are depicted in cartoon and surface representations, respectively. MNT$^{Lpg}$ is shown in violet; the HEPN$^{Lpg}$ tetramers are depicted in the same colors as in Fig. 1a. **a** Front view. The residues involved in the dimerization of HEPN$^{Lpg}$ with MNT$^{Lpg}$ are highlighted in darker colors (in Interface I, deep teal; in Interface II, green). **b** Opposite view. The corresponding residues from chain B and chain D are highlighted in darker colors (in chain B, light teal; in chain D, deep gray). **c** Dimerization interfaces in the HEPN$^{Lpg}$-MNT$^{Lpg}$ complex. The regions containing hydrophilic interactions from Interfaces I and II are represented. Hydrogen bonds and salt bridges are represented as dashed black and green lines, respectively, while both hydrogen bonds and salt bridges are depicted as dashed red lines. **d** Stoichiometric confirmation of HEPN$^{Lpg}$-MNT$^{Lpg}$ in solution. Size exclusion chromatography (SEC) demonstrates that MNT$^{Lpg}$ binds with HEPN$^{Lpg}$ in its tetrameric state, rather than in its dimeric state. The representative samples injected into the column are marked with different colors, while the result of the standard components is indicated with a dashed gray line.

## MNT$^{Lpg}$ modulates HEPN$^{Lpg}$ despite lacking the α4 helix

Previously, MNT homologs lacking the C-terminal α4 helix were shown not to inhibit the toxicity of the cognate HEPN toxin since they abolish complex formation and modification reactions[15,16]. Intriguingly, an MNT$^{Lpg}$ lacking the α4 helix is found within the same operon as its cognate HEPN$^{Lpg}$, forming a HEPN–MNT module. Through structural and biochemical analyses, the molecular mechanism linking HEPN$^{Lpg}$ and MNT$^{Lpg}$ was revealed. In the HEPN$^{Lpg}$-MNT$^{Lpg}$ complex structure, one MNT$^{Lpg}$ could only bind to tetrameric HEPN$^{Lpg}$ to form a dual binding interface, and the MNT-bound residues of HEPN$^{Lpg}$ in these interfaces are located near to each other. However, the snapshot of AMPylated HEPN$^{Lpg}$ revealed that the Y105 residues of all four chains were AMPylated. How could this process be accomplished? One explanation is that MNT$^{Lpg}$ first modifies HEPN$^{Lpg}$ in the dimeric state. Afterward, two homodimers of AMPylated HEPNs$^{Lpg}$ could assemble into a tetramer by hydrophilic interactions at tetramer interfaces. Unfortunately, a complex between MNT$^{Lpg}$ and the HEPN$^{Lpg}$ dimer could not be obtained. Meanwhile, HEPN$^{Lpg}$ cannot be AMPylated by its cognate MNT$^{Lpg}$ when it was in the dimeric state. These results indicate that the nucleotidyl transfer process between MNT$^{Lpg}$ and dimeric HEPN$^{Lpg}$ could be hindered. In either scenario, it could be speculated that MNT$^{Lpg}$ sequentially binds to the HEPN$^{Lpg}$ tetramer at the adjacent binding sites to induce a modification reaction. As shown in Fig. 8a, chains A and C of the HEPN$^{Lpg}$ tetramer could be initially AMPylated. Afterward, dissociation and reassociation of HEPN$^{Lpg}$ might be accomplished as a result of the transient and reversible self-assembly process. When the nonmodified chains of HEPN$^{Lpg}$ (here, chains B and chain D of HEPN$^{Lpg}$) reassemble to form adjacent binding sites, MNT$^{Lpg}$ can AMPylate Y105 in these chains in the same manner as in chains A and C.

A comparison of the three homologs from the HEPN–MNT system revealed that all the MNTs share a common binding interface between the α2 helix of MNT and HEPN. Notably, the additional α4 helix in MNT from *S. oneidensis* and *A. flos-aquae* provides a secondary docking site with HEPN (Supplementary Fig. 11b, c). Intriguingly, α2–β3 loop in MNT$^{Lpg}$ interacts with HEPN$^{Lpg}$ to form a secondary docking site (Fig. 8c and Supplementary Fig. 11a). This is why MNT$^{Lpg}$ can specifically bind to the HEPN$^{Lpg}$ tetramer. However, unmodified HEPN$^{Lpg}$ was detected via in vivo and in vitro AMPylation assays, which indicated that the efficiency of the MNT$^{Lpg}$ modification was relatively low. Presumably, the lack of an α4 helix in MNT$^{Lpg}$ impaired the binding affinity of MNT$^{Lpg}$ for HEPN$^{Lpg}$.

## Two types of mechanisms for controlling HEPN$^{Lpg}$ toxicity

In this study, we investigated the inhibitory mechanism of HEPN$^{Lpg}$. Based on structural and biophysical analyses, the dimer–tetramer

equilibrium in HEPN$^{Lpg}$ was measured. When two HEPN$^{Lpg}$ dimers are self-assembled into HEPN$^{Lpg}$ tetramer, the tetrameric interfaces consist of α2 and α3 helices and α2–α3 loops. Notably, the α2–α3 loop inserts into the deep cleft formed by the corresponding HEPN$^{Lpg}$ dimer (Fig. 3e). Moreover, the α2–α3 loop passes through the midst of the cleft comprising the active site, resulting in steric hindrance within the active site of HEPN$^{Lpg}$. On the other hand, HEPN$^{Lpg}$ toxicity can be inhibited via enzymatic modification by MNT$^{Lpg}$. When Y105 of HEPN$^{Lpg}$ was AMPylated, the AMP molecules were positioned near the RX$_4$HXY motif and formed several extensive interactions (Fig. 7b). Moreover, several residues from the N-terminus of HEPN$^{Lpg}$ form the positively charged crevices of HEPN$^{Lpg}$, which are positioned at the bilateral bases of the deep cleft. Presumably, these crevices might be sites where RNA can bind with HEPN$^{Lpg}$ through charge–charge interactions. Notably, the covalently bonded AMPs are positioned near the bilateral bases, which can interfere with substrate binding, resulting in inhibition of HEPN$^{Lpg}$ activity.

## Overall insight into the regulatory process of the HEPN$^{Lpg}$-MNT$^{Lpg}$ module

Finally, the overall regulatory process of the HEPN$^{Lpg}$-MNT$^{Lpg}$ module was revealed. Under suitable growth conditions, HEPN$^{Lpg}$ toxin is inhibited through several steps (Fig. 9). First, the HEPN$^{Lpg}$ dimer self-assembles to form tetrameric HEPN$^{Lpg}$, fulfilling two specific requirements. One is associated with complex formation between HEPN$^{Lpg}$ and MNT$^{Lpg}$. As mentioned above, if the α4 helix is absent, it is challenging for MNT$^{Lpg}$ to interact with dimeric HEPN$^{Lpg}$. Therefore, self-assembly of HEPN$^{Lpg}$ promotes the formation of the HEPN$^{Lpg}$-MNT$^{Lpg}$ complex. Moreover, it can mitigate the adverse impacts of decreased MNT$^{Lpg}$ modification activity. According to the results of the AMPylation assays, HEPN$^{Lpg}$ was not fully AMPylated by MNT$^{Lpg}$, which might be a disadvantage for *L. pneumophila* at the cellular level. However, self-assembly of HEPN$^{Lpg}$ triggers an auto-inhibitory mechanism that serves as a "secondary check" to regulate its toxicity. Moreover, this mechanism is distinct from the conditional cooperativity of the type II TA system, which relies on the stoichiometric relationship between the toxin and the antitoxin[29]. The subsequent process of complex formation can be divided into two pathways based on the ATP concentration. At low concentrations of ATP, HEPN$^{Lpg}$ cannot be AMPylated, and it becomes activated upon the degradation of labile MNT$^{Lpg}$. Alternatively, HEPN$^{Lpg}$ can be tri- or tetra-AMPylated at high concentrations of ATP, which abolishes the activity of HEPN$^{Lpg}$. Noticeably, the dissociation and subsequent reassociation of HEPN$^{Lpg}$ might be carried out to achieve complete AMPylation of all the chains within tetrameric HEPN$^{Lpg}$. Subsequently, AMPylated HEPN$^{Lpg}$ might be activated via de-AMPylation by phosphodiesterases. Finally, the freely released HEPN$^{Lpg}$ toxins from these two pathways act as a metal-dependent RNases to digest RNA in a non-specific manner, which can interfere with protein expression under stressful conditions. Taken together, the results revealed the regulatory mechanism of HEPN$^{Lpg}$-MNT$^{Lpg}$ at the molecular level, providing insight into the HEPN–MNT module.

## Methods
### Gene cloning and mutagenesis
A set of genes was identified within a single operon from *Legionella pneumophila* Philadelphia 1, specifically HEPN$^{Lpg}$ (*lpg2920*) and MNT$^{Lpg}$ (*lpg2921*), with an overlap of 8 base pairs observed between the genes of HEPN$^{Lpg}$ and MNT$^{Lpg}$. The genes encoding HEPN$^{Lpg}$, MNT$^{Lpg}$, and the HEPN$^{Lpg}$-MNT$^{Lpg}$ complex were synthesized by Bionics (Seoul, Korea) and amplified by polymerase chain reaction (PCR) using the following primers (Supplementary Table 3). The PCR products were digested by Nde I and Xho I and ligated separately into pET-28a (+), with N-terminal His-tags. The recombinant plasmids were subsequently transformed into the *E. coli* strain XL10-Gold for amplification.

To identify the residues that are essential for the activity of HEPN$^{Lpg}$ and MNT$^{Lpg}$, as well as the tetrameric interface for HEPN$^{Lpg}$, several mutations were introduced using the EZchang™ Site-Directed Mutagenesis Kit (Enzynomics, Daejon, Korea) according to the manufacturer's protocol. The mutations in HEPN$^{Lpg}$ were Q64AR98A, Q64AH103A, Q64AQ44A, Q64AE47A, Q64AQ44AE47A, Q64A and R73A, while the mutations in MNT$^{Lpg}$ were G36AS37T, D48E, D50E, and D48ED50E. The primers used for mutagenesis are described in Supplementary Table 3.

### Protein expression and purification
The recombinant HEPN$^{Lpg}$ WT protein and its mutants were over-expressed in *E. coli* Rosetta2 (DE3) cells using Luria–Bertani (LB) medium supplemented with kanamycin. *E. coli* Rosetta2 (DE3) cells harboring HEPN$^{Lpg}$ WT were grown until the OD$_{600}$ reached 0.5–0.6, after which the proteins were induced with 0.5 mM isopropyl β-D-1-thiogalactopyranoside (IPTG) for 4 h at 37 °C. Due to the high toxicity, the cells harboring HEPN$^{Lpg}$ Q64A and R73A were grown to the stationary phase (OD$_{600}$ of ~ 1–1.2) and induced by 0.5 mM IPTG for only 2 h; to ensure consistent induction conditions, all HEPN$^{Lpg}$ mutants potentially related to the active site (Q64AR98A, Q64AH103A, Q64AQ44A, Q64AE47A, and Q64AQ44AE47A) were induced in the same manner as HEPN$^{Lpg}$ Q64A. The cells were harvested by centrifugation at 6,400 × g for 10 min and frozen at −80 °C. The harvested cells were resuspended in buffer A (20 mM Tris-HCl, pH 7.9, and 500 mM NaCl) containing 10% (v/v) glycerol and lysed by ultrasonication. After centrifugation at 28,300 × g for 1 h at 10 °C, the supernatant containing the HEPN$^{Lpg}$ protein with an N-terminal His-tag was applied to a Ni$^{2+}$-NTA column (Qiagen, Hilden, Germany), pre-equilibrated with buffer A and washed with buffer A containing 50 mM imidazole. The target proteins were eluted with an imidazole gradient (100 mM–500 mM). The next purification step involved size-exclusion chromatography (SEC) using a Hiload 16/60 Superdex 200 prep-grade column (GE Healthcare, Chicago, IL, USA) preequilibrated with buffer containing 20 mM HEPES (pH 7.5) and 200 mM NaCl. For the standalone MNT$^{Lpg}$ and its mutants, the entire process of expression and purification steps was identical to that of HEPN$^{Lpg}$. To isolate NMPylated HEPN$^{Lpg}$, cells harboring the plasmid encoding the HEPN$^{Lpg}$-MNT$^{Lpg}$ complex were grown until the OD$_{600}$ reached 0.4–0.6 and induced with 0.2 mM IPTG for 20 h at 15 °C. After the cells were lysed by ultrasonication, the supernatant was loaded on a Ni$^{2+}$-NTA column and washed with buffer A containing 50 mM imidazole. The NMPylated HEPN$^{Lpg}$ proteins were eluted with an imidazole gradient ranging from 300 mM to 500 mM. In this scenario, most of the standalone MNT$^{Lpg}$ was eluted in the wash buffer. The second purification step involving SEC was conducted in the same manner as that used for HEPN$^{Lpg}$. To obtain the HEPN$^{Lpg}$-MNT$^{Lpg}$ complex, the purified HEPN$^{Lpg}$ WT protein was incubated with purified MNT$^{Lpg}$ for 30 min at 20 °C. The sample was then subjected to a Hiload 16/60 Superdex 200 prep-grade column of SEC for further purification. The purity of the target protein was confirmed by SDS–PAGE.

### Crystallization and diffraction data collection
Crystallization was performed using the sitting-drop vapor diffusion method at 20 °C. The initial screening was conducted by mixing equal volumes (0.5 μl each) of protein solution and reservoir solution, and the conditions were subsequently further optimized. The protein concentrations of HEPN$^{Lpg}$ WT, HEPN$^{Lpg}$ Q64A, and NMPylated HEPN$^{Lpg}$ used for crystallization were 13 mg/ml; the best crystal of HEPN$^{Lpg}$ WT was grown in the presence of 20% (w/v) PEG 3350 and 0.2 M ammonium tartrate dibasic (pH 6.5). The crystal buffer for HEPN$^{Lpg}$ Q64A was 20% (w/v) PEG 3350 and 0.2 M lithium acetate dehydrate. For NMPylated HEPN$^{Lpg}$, the crystal buffer consisted of 25% (w/v) PEG 3350, 0.1 M Bis-Tris and 0.2 M ammonium sulfate (pH 5.5). The MNT$^{Lpg}$ protein was concentrated to 3 mg/ml and crystallized in a buffer containing 20%

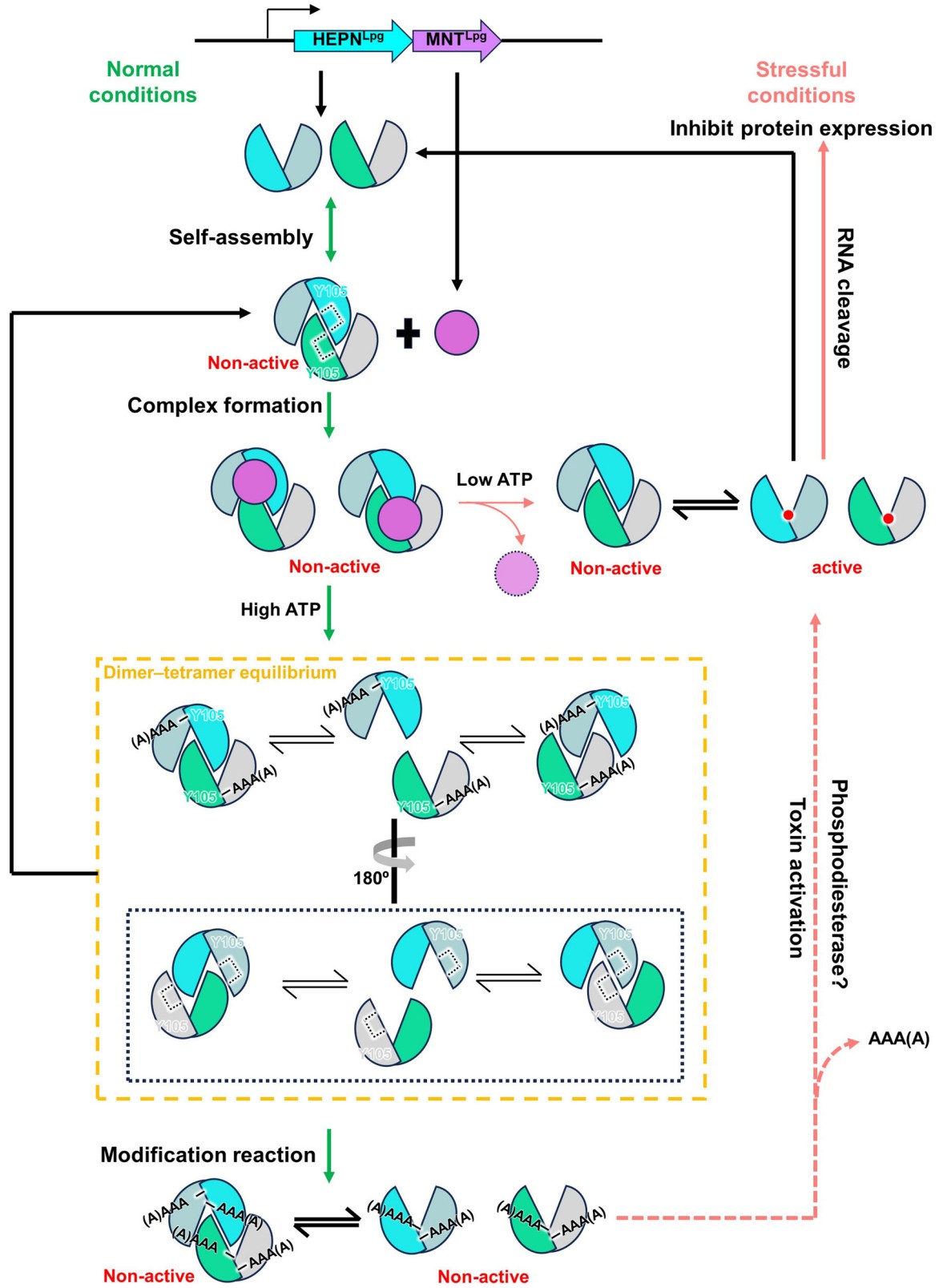

(w/v) PEG 6000, 0.1 M citric acid/sodium hydroxide and 1 M lithium chloride (pH 4.0). A crystal of the HEPN^Lpg-MNT^Lpg complex was obtained at a concentration of 12 mg/ml in the presence of 20% (w/v) PEG 3350 and 0.2 M ammonium citrate tribasic (pH 7.0). Prior to data collection, all of the optimized crystals were soaked in a cryoprotectant consisting of the reservoir solution with 20% (v/v) glycerol. The diffraction data were collected using a Quantum Q270r CCD detector (ADSC, Poway, CA, USA) at beamline 7 A and a PILATUS3 6 M CCD detector (Dectris, Baden-Daettwil, Switzerland) at beamline 11 C of the Pohang Light Source, Republic of Korea, as well as an EIGER X 4 M (x2) detector (Dectris) at beamline BL-1A of the Photon Factory, Japan.

The raw data were processed and scaled using the *HKL2000* program package[30] and XDS program package[31]. The detailed data statistics are summarized in Supplementary Table 4.

**Fig. 9 | A proposed model for the regulatory mechanism of the HEPN$^{Lpg}$-MNT$^{Lpg}$ module.** Under normal conditions, the HEPN$^{Lpg}$ dimer undergoes initial self-assembly to form a HEPN$^{Lpg}$ tetramer, resulting in autoinhibition of HEPN$^{Lpg}$ activity. Then, MNT$^{Lpg}$ binds to the HEPN$^{Lpg}$ tetramer through adjacent binding sites (black dashed squares). The subsequent processes diverge into two pathways depending on the concentration of ATP. In the low ATP pathway, HEPN$^{Lpg}$ remains unmodified by MNT$^{Lpg}$, and the separation of MNT$^{Lpg}$ from HEPN$^{Lpg}$ occurs due to the transient affinity between the two proteins. In the high ATP pathway, MNT$^{Lpg}$ can initially AMPylate two chains of the HEPN$^{Lpg}$ tetramer with adjacent binding sites. Moreover, the HEPN$^{Lpg}$ tetramer undergoes dissociation and reassociation, triggering complete AMPylation of HEPN$^{Lpg}$ (yellow dashed box). Under stressful conditions, AMPylated HEPN$^{Lpg}$ undergoes reactivation through the enzymatic digestion of AMP, facilitated by a putative phosphodiesterase. The active HEPN$^{Lpg}$ dimer from these two pathways cleaves RNA, inhibiting protein expression in *L. pneumophila*. The processes involved in normal conditions and stressful conditions are indicated by green and salmon arrows, respectively.

## Structure determination and refinement

The HEPN$^{Lpg}$ WT and MNT$^{Lpg}$ structures were solved by the molecular replacement (MR) method using the program Phaser-MR[32] with a prediction model from Alphafold2[33] as an initial template, and resolutions of ~1.8 Å and ~2.8 Å were obtained, respectively. The refined model of HEPN$^{Lpg}$ WT was subsequently used to solve the phase problem for NMPylated HEPN$^{Lpg}$ and HEPN$^{Lpg}$ Q64A using the same MR method at resolutions of ~1.7 Å and ~1.59 Å, respectively. The structure of the HEPN$^{Lpg}$-MNT$^{Lpg}$ complex was solved through Phaser-MR, utilizing refined HEPN$^{Lpg}$ WT and MNT$^{Lpg}$ as templates at a resolution of ~2.4 Å. Five percent of the data were randomly reserved as the test set to calculate $R_{free}$ for the entire dataset[34]. The model was manually modified using the program *Coot*[35] and automatically refined by *Refmac* in the *CCP4 Program suite*[36] and *phenix. refine* in *PHENIX*[37]. Water molecules and ligands were introduced using the program *Coot*[35]. All the final models were subjected to stereochemical analysis using MOLPROBITY[38]. PISA[39] and protein interaction calculator (*PIC*)[40] were used to calculate the interface area and associated interactions. *PyMOL* (PyMOL Molecular Graphics System, version 2.1; Schrödinger, LLC., Cambridge, MA, USA) was used to visualize and generate figures.

## Analytical ultracentrifugation (AUC)

Analytical ultracentrifugation was conducted using an Optima AUC (Beckman Coulter, Inc., Brea, CA, USA) equipped with an absorbance detector set to 280 nm. Sedimentation velocity experiments were performed at a rotor speed of 45,000 rpm with a 60Ti rotor. The experiments utilized a 2-sector EPON centerpiece, which was loaded with ~0.3 mg/ml (10 μM) HEPN$^{Lpg}$ (WT, Q64A, R73A) in a final buffer composed of 20 mM HEPES (pH 7.5) and 200 mM NaCl. The reference solution used was the final buffer alone. Data analysis was carried out using the SEDFIT software, version 16.1c.

## Surface plasmon resonance (SPR)

The homotetramerization affinities of HEPN$^{Lpg}$ WT, HEPN$^{Lpg}$ Q64A, and HEPN$^{Lpg}$ R73A were performed using SPR kinetics experiments. All experiments were conducted at 25 °C using a Biacore T200 system (GE Healthcare). Amine coupling was employed for immobilization, utilizing a kit containing 0.1 M N-hydroxysuccinimide and 0.4 M 1-ethyl-3-(3-dimethylaminopropyl) carbodiimide hydrochloride on a CM5 sensor chip. HBS-P buffer (10 mM HEPES, pH 7.5, 150 mM NaCl, and 0.005% Tween 20) was used according to the manufacturer's protocol (GE Healthcare). For homotetramerization affinity measurements, each HEPN$^{Lpg}$ WT, HEPN$^{Lpg}$ Q64A, and HEPN$^{Lpg}$ R73A, dissolved in 10 mM sodium acetate at pH 5.5, was injected and immobilized on the flow cell 2 (sample cell) at 10 μg/mL as ligands until the immobilization level reached approximately 600 response units (RU). To prevent redundant self-assembly of HEPN$^{Lpg}$ during immobilization, sufficient HBS-P buffer was applied to achieve the desired immobilization level. The remaining activated carboxyl groups on the CM5 sensor chip surface were deactivated with 1 M ethanolamine at pH 8.5 for 450 s. The control was settled identically with flow cell 1 (reference cell) without immobilized proteins to subtract the response from each sample dataset. The corresponding proteins of immobilized ligands (HEPN$^{Lpg}$ WT, HEPN$^{Lpg}$ Q64A, and HEPN$^{Lpg}$ R73A) were diluted in HBS-P buffer at concentrations of 0.39 μM, 0.78 μM, 1.56 μM, 3.13 μM, and 6.25 μM. They were injected as analytes at a rate of 10 μL/min for 120 s overflow cells 1 and 2, followed by dissociation for 300 s in multi-cycle reactions. As the response units adequately returned to baseline following dissociation, no regeneration process was performed. SPR response data were fit to a steady-state affinity model to determine the maximal response ($R_{max}$) using Biacore T200 evaluation software 3.0 (GE Healthcare).

## Molecular Dynamics (MD) simulation

The Maestro software from the Schrödinger suite was employed for the molecular dynamics simulations. Initially, the protein models for HEPN$^{Lpg}$ WT and HEPN$^{Lpg}$ Q64A, were prepared by the protein preparation wizard with OPLS4 force field[41]. Desmond was used to carry out the molecular dynamics simulations. For explicit solvent simulations, periodic boundary conditions with orthorhombic boxes buffered at $10 \times 10 \times 10$ Å distances were applied. The system was solvated using the TIP3P water model and supplemented with 150 mM NaCl after being neutralized with sodium or chloride ions to maintain an electrical balance. The solvated system containing protein underwent energy minimization and relaxation for 100 ps using the minimization step in Desmond with the OPS2005 forcefield. The simulations were conducted in the NPT ensemble (isothermal and isobaric simulations) using the Martyna-Tobias-Klein method for isotropic pressure, maintained at 1 atm and the Nose-Hoover thermostat algorithm for constant temperature at 300 K[42,43]. A total of 200 ns simulations were performed, with trajectories saved at 200 ps intervals. The trajectories were analyzed using the simulation interaction diagram. MD simulations input model and output files and MD simulations reliability and reproducibility checklist are available as Supplementary Data 1.

## Total RNA extraction from *L. pneumophila* and in vitro total RNA digestion assay

To isolate total RNA from *L. pneumophila* Philadelphia 1, the bacterial strain was obtained from the Korean Collection for Type Cultures (KCTC). The cells were cultured in buffered yeast extract (BYE) broth, which consists of 10 g yeast extract, 1 g α-ketoglutarate, 10 g ACES buffer, 0.4 g L-cysteine HCl, and 0.25 g iron (III) pyrophosphate per liter of purified water, with the pH adjusted to 6.9. When *L. pneumophila* cells were grown until the OD$_{600}$ reached 1–1.1, the cells were harvested by centrifugation at $18,000 \times g$ for 2 min. Total RNA, including rRNA and tRNA, was isolated from *L. pneumophila* using the TRI reagent RNA isolation method. To further isolate the small-sized RNA (5S rRNA and tRNA) from the purified total RNA, a FARB Mini Column (FAVORGEN, Ping Tung, Taiwan) was utilized. The 5S rRNA and tRNA were separated into the flow-through by centrifugation at $18,000 \times g$ for 30 sec. They were further purified using 75% ethanol and finally dissolved in RNase-free water.

To determine the RNase activity of dimeric HEPN$^{Lpg}$ toxin, an in vitro total RNA digestion assay was conducted with dimeric HEPN$^{Lpg}$ using total RNA from *L. pneumophila* at 37 °C for 30 min. The 10 μl reaction mixture contained 1 μM RNA with a dimeric HEPN$^{Lpg}$ concentration range of 10 nM–160 nM. The assays were conducted in buffer containing 20 mM HEPES (pH 7.5) and 200 mM NaCl. The reaction mixture was subjected to electrophoresis on a 1.8% agarose gel with 0.5x TBE for 25 min.

To verify whether dimeric HEPN$^{Lpg}$ can cleave rRNA in an intact ribosome, the purified 70S ribosome was obtained from New England

Biolabs (Ipswich, MA, USA). The ribosome was dissolved in a storage buffer (20 mM HEPES-KOH, pH 7.6, 10 mM Mg(OAc)$_2$, 30 mM KCl, and 7 mM beta-mercaptoethanol) to maintain its integrity. For the in vitro rRNA digestion assay, 1 μM ribosome was used, following the same method as the total RNA digestion assay. To improve the resolution for detecting small-sized RNA, a 12% polyacrylamide gel was used for electrophoresis to analyze the impact of HEPN$^{Lpg}$ on 5S rRNA and tRNA. The electrophoresis was conducted using 0.5x TBE at 100 V for 120 min. All electrophoresis results were visualized using a PrintGraph 2 M.

### In vitro RNase assay

To conduct a quantitative analysis of the RNase activity of apo HEPN$^{Lpg}$, a fluorescence quenching assay was conducted on dimeric HEPN$^{Lpg}$ and its variants using an RNase Alert Kit (IDT, Coralville, IA, USA). A synthetic RNA oligonucleotide covalently linked to fluorescein at one end and a dark quencher at the other end exhibited fluorescence quenching. When the RNA was digested by an RNase, fluorescein was liberated from the quencher. The released fluorescein exhibited green fluorescence (490 nm excitation, 520 nm emission), which was measured via RFU on a SpectraMax M5 plate reader (Molecular Devices, San Jose, CA, USA). The assays were performed in a buffer comprising 20 mM HEPES (pH 7.5) and 200 mM NaCl at 37 °C. To compare the RNase activity of dimeric HEPN$^{Lpg}$ with that of its variants, 20 μl of reaction mixture containing 2 μM protein and 0.2 μM RNA was used. The fluorescence endpoints were detected after a 1 h reaction. To assess the potential impact of contamination from other RNases in the in vitro RNase assays, we employed the Ribolock RNase inhibitor (henceforth, Ribolock) (Thermo Scientific, Waltham, MA, USA), which effectively inhibits the activity of various RNases. 1 U Ribolock was added to reaction mixture containing HEPN$^{Lpg}$ samples (including dimeric HEPN$^{Lpg}$ or its mutants) and RNA substrates. Fluorescence kinetics were detected at 30-s intervals throughout a 1 h reaction. To measure the Michaelis–Menten kinetics of dimeric HEPN$^{Lpg}$ and its variants, 20 μl of reaction mixture containing 0.35 μM protein and an RNA concentration ranging from 0.1–1.4 μM were used. The initial velocities and kinetic parameters $K_m$ and $k_{cat}$ were analyzed using *Prism* software (version 10.0.1 for macOS; GraphPad Software, Boston, Massachusetts, USA). All pipette tips and tubes used in the in vitro RNase assay were autoclaved, and the laboratory bench surfaces and pipettes were treated with RNaseZapTM RNase Decontamination Solution (Invitrogen, Carlsbad, CA, USA) to prevent RNase contamination during the assays.

### Spot-plating assays

Cultures of *E. coli* Rosetta2 (DE3) cells harboring the pET 28a vector with each HEPN$^{Lpg}$ variant were grown overnight at 37 °C in liquid LB media supplemented with kanamycin. For spot assays, the overnight cultures were diluted 1:100 and grown at 37 °C until the OD$_{600}$ reached ~0.5. The cells were harvested by centrifugation at 5000 rpm for 5 min, washed in phosphate-buffered saline (PBS), and serially diluted ($10^{-2}$–$10^{-6}$) before being spotted onto LB agar plates containing the indicated amount of inducer or repressor. Gene expression from plasmids carrying the pET promoter was repressed by 1% glucose and induced by a final concentration of 0.2 mM IPTG. LB agar plates were incubated at 37 °C for approximately 16 hr.

### Size-exclusion chromatography with multiangle light scattering (SEC-MALS)

SEC-MALS experiments were performed using an FPLC system (GE Healthcare) connected to a Wyatt MiniDAWN TREOS MALS instrument and an Optilab rEX differential refractometer (Wyatt, Santa Barbara, CA, USA). A Superdex 200 10/300 GL (GE Healthcare) gel-filtration column preequilibrated with buffer containing 20 mM HEPES (pH 7.5) and 200 mM NaCl was used. Purified MNT$^{Lpg}$ was injected at a

concentration of 3 mg/ml. The sample was injected at a flow rate of 0.4 mL/min. The data were analyzed using the Zimm model for fitting static light-scattering data and graphed using the EASI graph with an RI peak in the ASTRA VI (Wyatt).

### In vitro NMPylation assay

In vitro NMPylation assays were performed to investigate the function of MNT$^{Lpg}$. The experiments were divided into two steps, namely, storage and reaction procedures. The storage procedures were dependent on the purpose of the experiments and were performed in storage buffer (20 mM HEPES, pH 7.5, and 200 mM NaCl). First, to identify the types of nucleotides that MNT$^{Lpg}$ transfers to HEPN$^{Lpg}$, 40 μM MNT$^{Lpg}$ was incubated at 20 °C for 30 min with NTP (ATP, CTP, and GTP) at a concentration range of 10 μM–160 μM. Second, to determine the NMPylation active site of MNT$^{Lpg}$, 40 μM MNT$^{Lpg}$ or its mutants (G36AS37T, D48E, D50E, and D48ED50E) was incubated with 160 μM ATP at 20 °C for 30 min. Third, to assess the impact of metal ions on the NMPylation ability of MNT$^{Lpg}$, EDTA-treated MNT$^{Lpg}$ was used. EDTA-treated MNT$^{Lpg}$ (40 μM) was incubated with 160 μM ATP and 1 mM metal ions (Mg$^{2+}$, Zn$^{2+}$, Ca$^{2+}$, Co$^{2+}$, and Ni$^{2+}$) at 20 °C for 30 min. After the storage procedures, the subsequent reaction steps were conducted in reaction buffer (20 mM HEPES, pH 7.5, 200 mM NaCl, 5 mM DTT, and 10 mM MgCl$_2$). 80 μM HEPN$^{Lpg}$ was added to the reaction buffer and incubated at 37 °C for an additional 1 hr. In the case of the EDTA-treated MNT$^{Lpg}$ sample, MgCl$_2$ was excluded from the reaction buffer. SDS–PAGE loading buffer was added to stop the reaction, and the results were analyzed via SDS–PAGE.

### Liquid chromatography-mass spectrometry (LC-MS) analysis

To identify the NMP products that were cleaved from NMPylated HEPN$^{Lpg}$, LC-MS was performed. Purified NMPylated HEPN$^{Lpg}$ was treated with phosphodiesterase I (PDE I) at 37 °C for 1 hr. The small molecules were then collected by 3 kDa MWCO membranes for further analysis. In this experiment, a buffer without small molecules was used as the control, and both the control and small molecule samples were analyzed in duplicate. LC-MS was performed using electrospray ionization mass spectrometry (ESI-MS) with an integrated HPLC/ESI-MS system (InfinityLab LC/MSD iQ, Agilent Technologies, G6160A). The sample was injected into an InfinityLab Poroshell 120 EC-C18 column (2.1 × 50 mm, 2.7 μm) at a flow rate of 0.3 ml/min. Elution was performed using a linear gradient of solvent A (0.1% formic acid in water) and solvent B (0.1% formic acid in acetonitrile) as follows: 0–1 min, 1%–5% B; 1–2 min, 5%–10% B; 2–3 min, 10%–5% B; and 3–4 min, 5%–1% B. The ionization capillary voltage was adjusted to 3500 V, while the fragmentor was set to 110 V. The raw data obtained from LC-MS were processed using OpenLab CDS (version 2.8; Agilent Technologies, Santa Clara, CA, USA). The software was employed to identify the peaks based on retention times and *m/z* values. Any ambiguous peaks were further validated manually by analyzing their fragmentation patterns. For data analysis, signal intensities were normalized to the internal standard, and baseline corrections were applied. LC-MS raw data are available with this paper as Supplementary Data 2.

### High-resolution LC-MS/MS

The molecular weights of the protein samples (apo HEPN$^{Lpg}$ and NMPylated HEPN$^{Lpg}$) were analyzed by MS using a triple time of flight (TOF) 5600+ system (AB Sciex, Framingham, MA, USA), which was equipped with an Ultimate 3000 HPLC (Thermo Scientific, Waltham, MA, USA). The respective proteins were obtained following the procedure outlined in the previous section. Each sample was injected once into an INNO-P C4 column (2.0 mm × 50 mm, 5 μm) at a flow rate of 0.4 ml/min. The elution gradient for the separation of proteins was as follows: 0–0.5 min, 15% B; 0.5–11 min, 15%–100% B; 11–15 min, 100% B; 15–15.5 min, 100%–15% B; and 15.5–20 min, 15% B. The mobile phase consisted of solvent A (0.1% formic acid in water) and solvent B (0.1%

formic acid in acetonitrile). Data were collected in both MS1 and MS2 modes, using a full scan in positive ionization mode. Ionization was performed via electrospray ionization (ESI). The ion source pressures for both source 1 and source 2 were set at 50 psi, while the curtain gas pressure was maintained at 25 psi. The desolvation temperature was kept at 500 °C, and the ion spray voltage was set to 5.5 kV. Nitrogen was used as the carrier gas, and the declustering potential (DP) was set at 60 V. High-resolution mass spectra of the proteins were acquired over an $m/z$ range of 200–4000. The results were analyzed using the *Sciex OS* program (Sciex, Framingham, MA, USA). High-resolution LC-MS/MS raw data for both apo HEPN$^{Lpg}$ and NMPylated HEPN$^{Lpg}$ are available in this paper as Supplementary Data 3.

### Monitoring the stoichiometry of HEPN$^{Lpg}$ and MNT$^{Lpg}$
To monitor the stoichiometry of HEPN$^{Lpg}$ and MNT$^{Lpg}$, native PAGE was performed. MNT$^{Lpg}$ was incubated with tetrameric HEPN$^{Lpg}$ or dimeric HEPN$^{Lpg}$ with different ratios at 20 °C for 30 min. The samples were subsequently subjected to electrophoresis on a 15% polyacrylamide gel at 120 V for 100 min in running buffer (30 mM Tris, pH 8.8, and 200 mM glycine). The results were visualized using Coomassie blue staining.

To quantify the stoichiometry of HEPN$^{Lpg}$ and MNT$^{Lpg}$ in solution, size exclusion chromatography (SEC) experiments were performed using an FPLC system (GE Healthcare). A Superdex 200 10/300 GL gel-filtration column (GE Healthcare) preequilibrated with a buffer containing 20 mM HEPES (pH 7.5) and 200 mM NaCl was used. The representative protein samples were injected at a flow rate of 0.4 mL/min. Standard components were injected using the same buffer and flow rate.

### Statistics and reproducibility
In vitro total RNA digestion assays in Fig. 3b, In vivo expression tests in Fig. 6a, In vitro AMPylation assays in Fig. 6c–e were repeated two or three times independently with similar results.

### Reporting summary
Further information on research design is available in the Nature Portfolio Reporting Summary linked to this article.

## Data availability
Atomic coordinates and structural factors have been deposited in the Protein Data Bank (PDB) under accession codes: 8XEO (MNT$^{Lpg}$), 8XEM (HEPN$^{Lpg}$ WT), 8YUF (HEPN$^{Lpg}$ Q64A), 8XDJ (AMPylated HEPN$^{Lpg}$) and 8XEH (HEPN$^{Lpg}$-MNT$^{Lpg}$ complex). Source data are provided with this paper. Molecular dynamics simulations input and output files and liquid chromatography–mass spectrometry raw data files are available as Supplementary Data 1–3. Source data are provided with this paper.

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

## Acknowledgements

We thank the staff members of the beamlines, PLS-5C and PLS-11C of Pohang Accelerator Laboratory (Republic of Korea), BL-1A of Photon Factory (Japan), and BL44XU of Spring-8 (Japan) under the collaborative Research Program of Institute for Protein Research, Osaka University for assistance during X-ray diffraction experiments; staff members of NICEM (Republic of Korea) for analyzing the high-resolution LC–MS/MS data; staff members of MasterMediTech (Republic of Korea) for providing technical advice on LC–MS analysis; Jin-Ku Park of Central Instrument Center at Mokpo National University (Republic of Korea) for analyzing the AUC data; and Seok-Won Jang for providing technical advice on cell-based experiments. This research was supported by National Research Foundation of Korea (NRF) grants funded by the Korean government, Grant/Award Numbers: 2018R1A5A2024425, RS-2024-00458561, RS-2022-NR071830, and RS-2023-00218543.

## Author contributions

C.J., C.H.J., D.K., B.W.H., and B.L. conceived and designed the experiments. C.J., C.H.J., and H.W.K. conducted the experiments. C.J. and C.H.J. solved the crystal structures. C.J. performed the structural and biochemical analyses. C.H.J. and Y.C. carried out the mass spectrometry. J.M.K. performed the surface plasmon resonance experiment and molecular dynamics simulation. C.J., C.H.J., S.K., H.H.L., D.K., B.W.H., and B.L. analyzed the data. C.J. wrote the manuscript, and D.K., B.W.H., and B.L. edited it. D.K., B.W.H., and B.L. supervised the work.

## Competing interests

The authors declare no competing interests.
