## [Transparent Peer Review file · Nature Communications]

Structural insight into the distinct regulatory mechanism of the HEPN–MNT toxin–antitoxin system in *Legionella pneumophila*

Corresponding Author: Professor Bong-Jin Lee

Version 0:

Reviewer comments:

Reviewer #1

(Remarks to the Author)

Jin et al determined the structures of the HEPN, AMPylated HEPN, MNT, and the HEPN–MNT complex from *Legionella pneumophila*, and performed biochemical studies to propose a novel regulatory mechanism of the HEPN–MNT module. I have the following major concerns:

1. The authors stated HEPN has an oligomeric state transition according to their concentrations. Fig. 1B showed the tetramer formation at the concentration 3.3-12 mg/ml. In fact, it seems impossible for HEPN to exist at such a high concentration in vivo/in bacteria. So the tetramer may only be formed in vitro and is not biological relevant. Another question is that Q64/R73 have similar interactions in these interfaces in Fig. 1A (II-IV)? Why only one or two mutations have so remarkable effect? The oligomeric state should be confirmed by AUC or DLS.
2. In the RNA cleavage activity (Fig. 2B), only remarkable RNA cleavage was observed at 8 μ M HEPN. This concentration seems to be too high for a purified protein used in a RNA cleavage assay. Also, there is no positive control.
3. In Fig. 3A and Table S1, the RNase activity of Q44N, E47D and Q44NE47D is 40-50% of WT, this seems not a significant decrease. The authors should test the RNA cleavage activity of Q44N, E47D, or provide other evidence to confirm whether they are catalytic residues.
On the other hand, the authors showed the RX4HXY motif is not the catalytic site for Lpg by ribonuclease activity assay. The possible role of the conserved motif in Lpg should be discussed.
4. In this HEPN structure, R98 and H103 are distant from each other, and the authors conclude they can not cooperate with each other to constitute an active site. Is it possible that such distance is caused by the absence of RNA substrate? In recent RnlA structure (PDB 6Y2Q), the two RX4HXY motifs are also distant in a non-canonical HEPN dimer in an inactive resting state.
5. In the purification of NMPylated HEPN proteins, the authors stated "To isolate NMPylated HEPN, cells harboring the plasmid encoding the HEPN-MNT complex were grown..... The NMPylated HEPN proteins were eluted with an imidazole gradient ranging from 300 mM to 500 mM. In this scenario, most of the standalone MNT was eluted in wash buffer." How NMPylated HEPN and MNT were separated and why standalone MNT was eluted in wash buffer?
6. In HEPN-MNT complex structure, it is surprising that one MNT monomer interacts with only subunit A and C of one HEPN tetramer. Have the authors evaluated the oligomeric state of the complex by SEC or AUC, to confirm the status in crystal structure is consistent with that in solution?
The authors should mutate the interacting residues and determine the binding affinities of the mutants to confirm the structure.

Reviewer #2

(Remarks to the Author)

The manuscript by Jin et al. provides structural and functional characterization of the type VII toxin-antitoxin system HEPN-MNT from the pathogenic bacterium *L. pneumophila*. The main difference from the previously characterized HEPN-MNT systems is that the antitoxin MNT(Lpg) lacks the C-terminal helix α_4 , which is important for the HEPN-MNT interactions in the characterized systems from *S. oneidensis* and *A. flos-aquae* (Yao et al., 2015 and 2020; Songailiene et al, 2020). The study is interesting since it characterizes a novel variant of HEPN-MNT toxin-antitoxin system. In the manuscript 4 crystal structures of apo HEPN(Lpg) and MNT(Lpg), their complex and AMPylated HEPN(Lpg) were solved. It was shown that the shorter variant MNT(Lpg) can modify HEPN(Lpg) in vivo and in vitro by adding 3-4 AMPs to the conserved Y105 residue. Based on structural and functional studies the authors proposed an alternative novel HEPN(Lpg) active site and its activation mechanism.

The major concern is the conclusions about the novel active site of the HEPN(Lpg) domain and its RNA cleavage activity.

1) First, Discussion and the final mechanism in Fig. 8 explain that the tetrameric form of HEPN(Lpg) is inactive. This is consistent with the protein expression, where the overexpression of the WT HEPN(Lpg) toxin unexpectedly is non-toxic (see Methods). Differently, overexpression of the dimeric mutants is toxic, again suggesting that the active form is a dimer. However, the conclusion about a non-functional conserved HEPN RXH (R98-H103, hereafter RH) active site is made based on the inactive tetrameric structure. The RH residues are located on the loop therefore they can change their conformation in the active dimer structure. But the structure of the dimeric HEPN(Lpg) variant is not available, therefore the conformation of the HR loop in the active state is unknown.

2) WT HEPN(Lpg) domain exists in a tetramer-dimer equilibrium, at lower concentrations it is a dimer (Fig. 1B), then one would expect that at higher concentrations HEPN(Lpg) to be inactive as a tetramer, and active at lower concentrations as a dimer. However, total RNA cleavage results are the opposite: at lower concentrations (more dimeric form) HEPN(Lpg) is inactive, the RNA cleavage activity is only observed at high micromolar protein concentrations, where HEPN(Lpg) is supposed to be inactive tetramer (Fig. 2B). Furthermore, mutations of the predicted novel active site only moderately reduce the observed cleavage activity (Fig. 3A). Mutational analysis of an alternative active site does not prove that the observed activity is related with the proposed active site, all cleavage data may be related to contaminant RNase activity. To prove the activity of HEPN(Lpg), I suggest to perform all cleavage experiments using dimeric mutant Q64A R73A, whose expression is toxic indicating its activity. The active site mutants, both of RH and of novel predicted active sites, should also be made in the dimeric mutant, that oligomerization does not affect protein activity. For the novel active site alanine mutations should be made, since changing into similar amino acids (Q44N, E47D) did not give clear answer about their importance.

3) To test the specificity of RNA cleavage, total RNA from *E. coli* could be used for comparison. Could HEPN(Lpg) also cleave mRNA as *S. oneidensis* HEPN (Yao et al, 2015) or tRNA as *A. flos-aquae* HEPN (Songailiene et al, 2020)? The resolution of the gel in Fig. 2B is not sufficient, if only a few nucleotides are cleaved from tRNA, as observed for *A. flos-aquae* HEPN. If the target is ribosomal RNA, can it be cleaved in ribosomes, as they exist in the cell?

To monitor the stoichiometry of HEPN(Lpg) and MNT(Lpg), native PAGE was performed mixing MNT(Lpg) with tetrameric HEPN(Lpg) or dimeric HEPN(Lpg): it is not clear which dimeric mutant was used. Native PAGE suggests that a fairly stable complex is formed between HEPN(Lpg) and MNT(Lpg), but the resolution of this method is limited. Therefore I propose to use SEC-MALS to characterize HEPN(Lpg) WT and its dimeric mutant (Q64A R73A) binding to MNT(Lpg) using different protein ratios. In vitro AMPylation experiment of HEPN(Lpg) dimeric mutant Q64A R73A could directly show that only the HEPN(Lpg) tetramer is AMPylated.

Reviewer #3

(Remarks to the Author)

This review of the manuscript by Jin and Jeon et al. focuses solely on the structural work presented. The structures appear to be of high quality and there are no apparent anomalies in the data collection or refinement statistics (Supp Table 3). It is also appreciated that a 2Fo-Fc omit map is shown for the AMPylated HEPN structure. Below are additional minor edits the authors should consider.

-it would be helpful to also report the buried surface area as a percentage of the entire surface area in addition to the value in square Angstroms

-All structure figures: Cyan and pale cyan lettering is very hard to read on a computer

-Supplementary figure 2 (and Discussion): Can the authors state whether or not there are any crystal contacts or other structural features that may influence the angle between helices α_2 of the cleft?

Version 1:

Reviewer comments:

Reviewer #1

(Remarks to the Author)

The authors have addressed most of my concerns. I have one question as follows:

Fig3b , In vitro total RNA digestion assay. The length in each band in the marker should be labelled. Can the total RNA

completely cleaved by RNase A? No cleavage products can be seen in the gel.

Reviewer #2

(Remarks to the Author)

The additional data included after the revision support the structural part of the manuscript. However, I am still concerned about the introduction of the new type active site in the well characterized HEPN fold. This is an important statement and since RNases are common contaminants in protein preparations, the RNase activity of the proposed new type active site should be double checked.

1. New data on the RNA cleavage by the dimeric versions of the Q44A, E47A, Q44AE47A mutants support their importance for the observed RNA cleavage activity. However, it is not clear from the methods whether or not identical expression/purification protocols were used for the Q64A and these mutants of the new type active site. It is now stated that: p.19: „The recombinant HEPNLpg WT protein and its mutants were overexpressed in *E. coli* 13 Rosetta2 (DE3) cells using Luria–Bertani (LB) medium supplemented with kanamycin. *E. coli* Rosetta2 (DE3) cells harboring HEPNLpg WT and its mutants (except for Q64A and R73A) were grown until the OD600 reached 0.5–0.6, after which the proteins were induced with 0.5 mM isopropyl β -D-1-thiogalactopyranoside (IPTG) for 4 h at 37°C. Due to the high toxicity, the cells harboring HEPNLpg Q64A and R73A were grown to the stationary phase (OD600 of ~ 1–1.2) and induced by 0.5 mM IPTG for only 2 h.“

If, as stated, the Q64A and Q64A+Q44A, Q64A+E47A, Q64A+Q44AE47A mutants were expressed under different conditions, possible contamination in the Q64A and R73A protein preparations with RNases (detected by the RNase Alert Kit used to compare RNase activity) cannot be excluded.

2. Substrate specificity. In the final schema in Fig. 9 the authors claim that HEPNLpg cleaves only ribosomal 16S and 23S rRNAs. However, HEPNLpg also cleaves synthetic RNA substrate used for the mutant activity assay and they did not test mRNA cleavage, so this is not correct.

Methods, In vitro RNase assay, p.25 RNase Alert Kit, not rRNA, was used for mutant activity characterization: „To conduct a quantitative analysis of the RNase activity of apo HEPNLpg, a fluorescence quenching assay was conducted on dimeric HEPNLpg and its variants using an RNase Alert Kit (IDT, Coralville, IA, USA)“. This kit is designed to test to test RNase contaminations and the supplier states that „The sequence of the RNaseAlert™ Substrate has been carefully optimized to detect several RNases, including RNase A, RNase T1, RNase I, micrococcal nuclease, S1 nuclease, mung bean nuclease, and Benzonase™“ (from <https://www.thermofisher.com/order/catalog/product/AM1964>). The cleavage of this substrate by the Q64A mutant indicates that HEPN is not specific for ribosomal RNA and can also cleave other RNA substrates. Thus, the activity observed in Fig. 3b and Fig. S3 is a non-specific RNase activity (no specific cleavage pattern is observed and all rRNA is degraded). However, the authors did not test mRNA cleavage, so this remains to be elucidated.

Therefore „23S rRNA & 16S rRNA cleavage“ on the right side of the schema (Fig. 9) should be changed to „RNA cleavage“.

3. Testing of the in vivo toxicity of both canonical and new active site mutants (Q64A, Q64A+R98A, Q64A+H103A, Q64A+Q44A, Q64A+E47A) is required as an alternative method and would definitely support new type active site hypothesis. Since Q64A is toxic, mutations in the correct active site should abolish its toxicity. This can be done by spot-plating assay comparing the growth of serial dilutions of the *E. coli* cells expressing mutants in the presence and absence of IPTG (+1% glucose in the absence of IPTG to prevent background expression from T7 promoter) (see in doi.org/10.7554/eLife.98528.1).

Minor comments:

1. p.17: Not all mutants are listed in the methods, double mutants (like Q44A+Q64A, E47A+Q64A, etc) are not: „The mutations in HEPNLpg were R98A, H103A, Q44A, E47A, Q44AE47D, Q64A and R73A, while the mutations in MNTLpg were G36AS37T, D48E, D50E, and D48ED50E.“

Q44AE47D mutant should be Q44AE47A.

2. Fig. S3: no units for the enzyme activity (y axis) and substrate concentration (x axis) are shown.

Reviewer #3

(Remarks to the Author)

The authors sufficiently addressed the initial review comments. The revised manuscript is acceptable in its current form.

Version 2:

Reviewer comments:

Reviewer #2

(Remarks to the Author)

Clarification of the mutant purification protocols and addition of in vivo data is in line with the proposed hypothesis of a novel active site of HEPN domain.

Still, I am confused about the interpretation of the RNA cleavage activity results. In the response to point 1, the authors state: „As a result, the RNA substrates were rapidly digested by RNase A (almost completely digested at the beginning of the

assay), whereas HEPN Lpg Q64A and R73A moderately cleaved the RNA substrates over time (as illustrated in the accompanying figure). This indicates that the RNase activities detected for HEPNLpg Q64A and R73A were due to inherent properties of those samples, not influenced by other RNases. Therefore, we can exclude the possibility of contamination by other RNases.”

The activity of the preparation depends on the amount of the active enzyme in it. The RNase A preparation contains a significant amount of active RNase A and hypothetically HEPN Lpg preparation may contain only traces of RNase A (presume that HEPN Lpg is inactive on the selected substrates). Theoretically, incubation of the same volume of preparations containing different amounts of the active enzyme over the time will result in different cleavage profiles: fast cleavage (strong signal) in the case of RNase A preparation and slow cleavage (weak signal) by contaminated RNase in the HEPN Lpg preparation. If the RNase A preparation would be diluted, e.g. 10-100-1000x, it could result in the similar cleavage profile as the HEPN Lpg preparation. In this experiment, shown in in the accompanying figure, it is impossible to exclude contamination by other RNases. Only loss of RNase activity by the active site mutants of HEPN Lpg can exclude contamination.

Point-by-point response to the referee's comments

Reviewer(s)' Comments to Author:

Reviewer #1:

Comments to the Author

Jin et al determined the structures of the HEPN, AMPylated HEPN, MNT, and the HEPN–MNT complex from *Legionella pneumophila*, and performed biochemical studies to propose a novel regulatory mechanism of the HEPN–MNT module. I have the following major concerns:

1. The authors stated HEPN has an oligomeric state transition according to their concentrations.

Fig. 1B showed the tetramer formation at the concentration 3.3-12 mg/ml. In fact, it seems impossible for HEPN to exist at such a high concentration in vivo/in bacteria. So the tetramer may only be formed in vitro and is not biological relevant.

Another question is that Q64/R73 have similar interactions in these interfaces in Fig. 1A (II-IV)? Why only one or two mutations have so remarkable effect? The oligomeric state should be confirmed by AUC or DLS.

Answer: Thank you for the insightful comments. In our previous study, the relative scales for UV absorption and light scattering could not be detected at relatively low concentrations of the target protein. Therefore, we injected protein samples varying in concentration from 0.74 mg/ml to 12 mg/ml to detect the oligomeric states of HEPN^{L-pg}. Indeed, as the reviewer noted, concentrations of 3.3-12 mg/ml HEPN^{L-pg} seem relatively high for bacteria. Thus, we have removed the SEC-MALS data in the Results section titled “Overall structure of HEPN from *L. pneumophila*”. Alternatively, we investigated the homotetramerization (self-assembly) process of HEPN^{L-pg} through the following methods.

First, as recommended by the reviewer, we conducted analytical ultracentrifugation (AUC) experiments to determine the oligomeric states of HEPN^{L-pg} in solution. First, we used concentrations of 10 nM (0.00032 mg/ml), 100 nM (0.0032 mg/ml), 1 μM (0.032 mg/ml), 10 μM (0.32 mg/ml), and 100 μM (3.2 mg/ml) HEPN^{L-pg}. However, both the absorbance (280 nm) and interference signals could not be precisely determined due to the detection limitations associated with target protein concentrations, which was below 10 μM. Thus, we ultimately chose an optimal concentration of 10 μM for HEPN^{L-pg} WT, HEPN^{L-pg} Q64A, and HEPN^{L-pg} R73A to conduct the AUC experiments. The results showed that HEPN^{L-pg} WT exists in

dimer–tetramer equilibrium in solution, while only the dimeric state was detected in the HEPN^{Lpg} Q64A and R73A samples.

Second, to further quantify the homotetramerization of HEPN^{Lpg}, we performed surface plasmon resonance (SPR) experiments using HEPN^{Lpg} WT, HEPN^{Lpg} Q64A, and HEPN^{Lpg} R73A. Both the binding kinetics and affinity of HEPN^{Lpg} tetramerization were validated. The results indicated that the tetramerization process exhibited considerable fast association and dissociation rates, suggesting that the self-assembly process is transient and reversible. Furthermore, HEPN^{Lpg} WT showed a greater tendency for self-assembly than HEPN^{Lpg} Q64A and HEPN^{Lpg} R73A, which explains why only the dimeric form of HEPN^{Lpg} existed in these mutants. Accordingly, we stated the result in the Results section titled “Overall structure of HEPN from *L. pneumophila*” (lines 24–25 of p. 5 and lines 1–14 of p. 6); and added new figures (Figure 2a–i).

Figure 2

Furthermore, a comparison of the interfacial areas revealed that the tetrameric interfaces are relatively smaller than the dimeric interfaces (874–882 Å² for dimeric interfaces vs. 311–419 Å² for tetrameric interfaces) (Supplementary Table 1). Additionally, only hydrogen bonds exist in these tetrameric interfaces, with the residues Q64 and R73 being involved in most of the interactions (eleven H-bonds out of twelve). Therefore, it is suggested that these two residues are crucial for the tetramerization of HEPN^{Lpg}.

2. In the RNA cleavage activity (Fig. 2B), only remarkable RNA cleavage was observed at 8 μM HEPN. This concentration seems to be too high for a purified protein used in a RNA cleavage assay. Also, there is no positive control.

Answer: We appreciate the reviewer’s insightful comment. Indeed, while many other toxin proteins from TA systems have demonstrated in vitro RNase activity at micromolar concentrations, we considered these concentrations to be relatively high in a cellular context. Therefore, we conducted RNA cleavage assays using HEPN^{Lpg} at concentrations ranging from 10 nM to 160 nM. Additionally, we included 1 unit of RNase A as a positive control.

In this context, it is noteworthy that HEPN^{Lpg} Q64A mutant was used in place of HEPN^{Lpg} WT for the RNase assays for the following reasons. First, as previously mentioned, HEPN^{Lpg} WT exists in both dimeric and tetrameric states in solution, and the tetrameric state, which interferes with the RNase activity of HEPN^{Lpg}, could be partially present due to the dimer–tetramer equilibrium. Therefore, dimeric HEPN^{Lpg} mutants (Q64A or R73A), which do not affect the active site, were used to eliminate tetramerization-mediated impairment of HEPN^{Lpg} RNase activity. Second, the Q64A mutant was preferred over the R73A mutant because the R73 residue is crucial for interacting with the cognate MNT^{Lpg}, potentially affecting the binding affinity between HEPN^{Lpg} and MNT^{Lpg}. In this regard, the dimeric HEPN^{Lpg} mutant used in subsequent assays was based on HEPN^{Lpg} Q64A to focus exclusively on the impact of homotetramerization.

As a result, the dimeric HEPN^{Lpg} effectively digested 23S rRNA and 16S rRNA at lower concentrations (80–160 nM), indicating that homotetramerization indeed impaired the RNase activity of HEPN^{Lpg} WT.

Accordingly, we revised the results in the Results section titled “HEPN^{Lpg} possesses a new catalytic site”; and replaced Figure 3b with the new version below.

3. In Fig. 3A and Table S1, the RNase activity of Q44N, E47D and Q44NE47D is 40-50% of WT, this seems not a significant decrease. The authors should test the RNA cleavage activity of Q44N, E47D, or provide other evidence to confirm whether they are catalytic residues.

On the other hand, the authors showed the RX4HXY motif is not the catalytic site for Lpg by ribonuclease activity assay. The possible role of the conserved motif in Lpg should be discussed.

Answer: Thank you for pointing this out. To date, all HEPN^{Lpg} mutants have been redesigned based on the dimeric HEPN^{Lpg} mutant (HEPN^{Lpg} Q64A) for the reason outlined in #2.

Additionally, substitution with similar amino acids (Q44N and E47D) did not clarify their importance to the RNase activity of HEPN^{Lpg}. Therefore, we utilized alanine mutants (Q44A, E47A, and Q44AE47A) to conduct in vitro RNase assays. The results indicated that the dimeric HEPN^{Lpg} Q44A, E47A, and Q44AE47A mutants exhibited significant decreases in RNase activity compared to the dimeric HEPN^{Lpg}. In contrast, the dimeric HEPN^{Lpg} R98A and H103A mutants had negligible impacts on RNase activity.

Furthermore, the removal of metal ions nearly abolished the RNase activity of dimeric HEPN^{Lpg}, with Mg²⁺ being the sole metal ion capable of rescuing the RNase activity of EDTA-treated dimeric HEPN^{Lpg}. The catalytic properties of HEPN^{Lpg} were further assessed using dimeric HEPN^{Lpg} and its variants. The results showed that the k_{cat}/K_m values significantly decreased in the dimeric HEPN^{Lpg} mutants (Q44A, E47A, and Q44AE47A) and in the EDTA-treated dimeric HEPN^{Lpg} compared to those in the dimeric HEPN^{Lpg}. These findings led us to conclude that the Mg²⁺-coordinated site is the active site of the HEPN^{Lpg} ribonuclease.

Additionally, since residues R98 and H103 of HEPN^{Lpg} did not contribute to its RNase activity, we hypothesize that these polar residues within this conserved motif might be involved in interactions with ATP substrates during the nucleotidyl transfer process.

Accordingly, we revised the result in the Results section titled "HEPN^{Lpg} possesses a new catalytic site" (third paragraph) and the Discussion section titled "HEPN^{Lpg} acts as a Mg²⁺-dependent ribonuclease" (lines 3–6 of p. 14); Additionally, we replaced Figure 4a, b and Supplementary Figure 3 with the new versions below.

Figure 4

Supplementary Figure 3

4. In this HEPN structure, R98 and H103 are distant from each other, and the authors conclude they can not cooperate with each other to constitute an active site. Is it possible that such distance is caused by the absence of RNA substrate? In recent RnIA structure (PDB 6Y2Q), the two RX₄HXY motifs are also distant in a non-canonical HEPN dimer in an inactive resting state.

Answer: Thank you for bringing up this point. Indeed, two distinct states were identified in the RnIA structure: the resting HEPN dimer (nonactive form) and the canonical HEPN dimer (active form).

In the resting state, the residues R318 and H323 of the RX₄HXY motif from cognate chains are spatially separated, thereby preventing the formation of an active site. Conversely, in the canonical dimeric state, HEPN adopts a V-shaped conformation

typical of active HEPN dimers observed across various subfamilies. Specifically, in this active state, R318 and H323 are positioned adjacent to each other and posed for catalysis, with the distance between N ϵ 2 of H323 and NH2 of R318 ranging from approximately 3.3 to 3.9 Å, while the distance between N ϵ 2 of H323 in separate subunits was approximately 7.1 Å (as illustrated in the accompanying figure).

In contrast, biochemical assays and mutagenesis experiments on HEPN^{Lpg} suggested that the corresponding residues R98 and H103 in its Y-loop (RH₄HXY motif) do not cooperate to form an active site. Specifically, the Y-loop region (RH₄HXY motif) does not undergo significant structural changes between the dimeric state (active state) and tetrameric state (nonactive state) of HEPN^{Lpg}, with residues R98 and H103 aligning closely with each other. Moreover, molecular dynamics (MD) simulations further indicate that the Y-loop in HEPN^{Lpg} is relatively rigid and less dynamic. Consequently, it is suggested that R98 and H103 in the Y-loop might not undergo significant conformational changes during RNA cleavage processes. Collectively, these observations imply that the RX₄HXY motif is unlikely to function as a catalytic site in HEPN^{Lpg} due to the specific structural characteristics of these highly conserved residues.

Accordingly, we stated the results in the Results section titled “HEPN^{Lpg} possesses a new catalytic site” (lines 23–25 of p. 7 and lines 1–10 of p. 8); and added new figures (Supplementary Figure 2).

Supplementary Figure 2

5. In the purification of NMPylated HEPN proteins, the authors stated “To isolate NMPylated HEPN, cells harboring the plasmid encoding the HEPN-MNT complex were grown..... The NMPylated HEPN proteins were eluted with an imidazole gradient ranging from 300 mM to 500 mM. In this scenario, most of the standalone MNT was eluted in wash buffer.”

How NMPylated HEPN and MNT were separated and why standalone MNT was

eluted in wash buffer?

Answer: Thank you for bringing up point. The results of the IMAC experiment involving two sample combinations, namely HEPN^{Lpg} (His-tagged)/MNT^{Lpg} (no tag) and HEPN^{Lpg} (no tag)/MNT^{Lpg} (His-tagged), are provided (as illustrated in the accompanying figure).

As outlined in the Methods section, proteins lacking His-tags were predominantly eluted during the loading and initial washing steps with 50 mM imidazole. Subsequently, during the elution phase with an imidazole gradient ranging from 100–200 mM, the HEPN^{Lpg}–MNT^{Lpg} complex began to elute, while proteins with a His-tag were eluted at higher imidazole concentrations (300–500 mM). This phenomenon can be attributed to two primary factors:

First, as previously mentioned, the HEPN–MNT modules are expected to engage in transient interactions for enzymatic modification, that are distinct from canonical type II TA systems. Additionally, in the case of MNT^{Lpg}, the absence of a long C-terminal $\alpha 4$ helix is expected to weaken its binding to HEPN^{Lpg}. Consequently, the protein (lacking a His-tag) that eluted during the wash phase appeared to either nonspecifically adhere to the Ni²⁺-NTA matrix or exhibit weak interactions with its His-tagged counterpart protein.

Second, the binding affinity may also be influenced by the composition of buffer A (20 mM Tris-HCl, pH 7.9, and 500 mM NaCl) utilized in the IMAC procedure. As noted previously, MNT^{Lpg} and HEPN^{Lpg} primarily interact through hydrogen bonds and salt bridges at interfaces I and II. Therefore, the high NaCl concentration in buffer A increases the ionic strength of the solution, potentially shielding these electrostatic interactions. This shielding could disrupt salt bridges, thereby weakening the interaction between protein components.

6. In HEPN-MNT complex structure, it is surprising that one MNT monomer interacts with only subunit A and C of one HEPN tetramer. Have the authors evaluated the oligomeric state of the complex by SEC or AUC, to confirm the status in crystal structure is consistent with that in solution?

The authors should mutate the interacting residues and determine the binding affinities of the mutants to confirm the structure.

Answer: Thank you for the comments. As suggested, we performed size exclusion chromatography (SEC) to verify whether the presence of the HEPN^{Lpg}-MNT^{Lpg} complex in the crystal structure corresponds to its solution state. Our findings indicate that upon the addition of MNT^{Lpg} to tetrameric HEPN^{Lpg}, HEPN^{Lpg}-MNT^{Lpg} complexes were indeed formed. In contrast, dimeric HEPN^{Lpg} (Q64A) did not interact with MNT^{Lpg} to form such complexes. Additionally, the SEC results were consistent with those obtained from native-PAGE.

Accordingly, we stated the result in the Results section titled “Crystal structural insight into the MNT^{Lpg}-HEPN^{Lpg} complex” (lines 12–15 of p. 13); and added new figures (Figure 8d).

Figure 8

d

As previously mentioned, HEPN^{Lpg} and MNT^{Lpg} engage through two interfaces (Interface I: 512 Å² and Interface II: 629 Å²) involving multiple residues that form hydrogen bonds and salt bridges. Specifically, R73 of HEPN^{Lpg} plays a crucial role by contributing extensively to strong hydrophilic interactions with MNT^{Lpg}. However, R73 also participates in the homotetramerization of HEPN^{Lpg}. Therefore, mutating R73 to alanine (R73A) not only impacts the binding between HEPN^{Lpg} and MNT^{Lpg} but also

affects the self-assembly process of HEPN^{Lpg} itself. Consequently, exclusively attributing the effects of the binding affinity of key residues between HEPN^{Lpg} and MNT^{Lpg} was challenging.

Reviewer #2:

Comments to the Author

The manuscript by Jin et al. provides structural and functional characterization of the type VII toxin-antitoxin system HENP-MNT from the pathogenic bacterium *L. pneumophila*. The main difference from the previously characterized HEPN-MNT systems is that the antitoxin MNT(Lpg) lacks the C-terminal helix alpha4, which is important for the HEPN-MNT interactions in the characterized systems from *S. oneidensis* and *A. flos-aquae* (Yao et al., 2015 and 2020; Songailiene et al, 2020). The study is interesting since it characterizes a novel variant of HEPN-MNT toxin-antitoxin system.

In the manuscript 4 crystal structures of apo HEPN(Lpg) and MNT(Lpg), their complex and AMPylated HEPN(Lpg) were solved. It was shown that the shorter variant MNT(Lpg) can modify HEPN(Lpg) in vivo and in vitro by adding 3-4 AMPs to the conserved Y105 residue. Based on structural and functional studies the authors proposed an alternative novel HEPN(Lpg) active site and its activation mechanism. The major concern is the conclusions about the novel active site of the HEPN(Lpg) domain and its RNA cleavage activity.

1) First, Discussion and the final mechanism in Fig. 8 explain that the tetrameric form of HEPN(Lpg) is inactive. This is consistent with the protein expression, where the overexpression of the WT HEPN(Lpg) toxin unexpectedly is non-toxic (see Methods). Differently, overexpression of the dimeric mutants is toxic, again suggesting that the active form is a dimer. However, the conclusion about a non-functional conserved HEPN RXH (R98-H103, hereafter RH) active site is made based on the inactive tetrameric structure. The RH residues are located on the loop therefore they can change their conformation in the active dimer structure. But the structure of the dimeric HEPN(Lpg) variant is not available, therefore the conformation of the HR loop in the active state is unknown.

Answer: Thank you for the suggestions. As recommended, we further elucidated the tertiary structure of dimeric HEPN^{Lpg} using X-ray crystallography, which represents an active form within the HEPN^{Lpg}-MNT^{Lpg} module. Notably, the Q64 residue and R73 residue play pivotal roles in homotetramerization. However, R73 also serves as a key residue involved in interacting with the cognate MNT^{Lpg}, suggesting its influence on binding between HEPN^{Lpg} and MNT^{Lpg}. Consequently, all dimeric

HEPN^{Lpg} mutants used in the structural analysis and subsequent assays were based on HEPN^{Lpg} Q64A to focus exclusively on the impact of homotetramerization.

Subsequently, our findings revealed that the Y-loop region (RH₄HXY motif) does not undergo significant structural changes between the dimeric state (active state) and the tetrameric state (nonactive state) of HEPN^{Lpg}, with residues R98 and H103 aligning closely with each other. Molecular dynamics (MD) simulations further indicate that the Y-loop in HEPN^{Lpg} is relatively rigid and less dynamic. Collectively, these observations confirm the conformation of the HR loop in its active state and suggest that the RX₄HXY motif is unlikely to function as a catalytic site in HEPN^{Lpg} due to the specific structural characteristics of these highly conserved residues.

Accordingly, we revised and stated the results in the Results section titled “HEPN^{Lpg} possesses a new catalytic site” (second paragraph); Additionally, we replaced Figure 3c, d and Supplementary Figure 2 with the new versions below.

Figure 3

Supplementary Figure 2

2) WT HEPN(Lpg) domain exists in a tetramer-dimer equilibrium, at lower concentrations it is a dimer (Fig. 1B), then one would expect that at higher concentrations HEPN(Lpg) to be inactive as a tetramer, and active at lower concentrations as a dimer. However, total RNA cleavage results are the opposite: at lower concentrations (more dimeric form) HEPN(Lpg) is inactive, the RNA cleavage activity is only observed at high micromolar protein concentrations, where HEPN(Lpg) is supposed to be inactive tetramer (Fig. 2B). Furthermore, mutations of

the predicted novel active site only moderately reduce the observed cleavage activity (Fig. 3A). Mutational analysis of an alternative active site does not prove that the observed activity is related with the proposed active site, all cleavage data may be related to contaminant RNase activity. To prove the activity of HEPN(Lpg), I suggest to perform all cleavage experiments using dimeric mutant Q64A R73A, whose expression is toxic indicating its activity. The active site mutants, both of RH and of novel predicted active sites, should also be made in the dimeric mutant, that oligomerization does not affect protein activity. For the novel active site alanine mutations should be made, since changing into similar amino acids (Q44N, E47D) did not give clear answer about their importance.

Answer: Thank you for the valuable suggestions. In our previous study, we considered that HEPN^{Lpg} predominantly exists in a dimeric state at a concentration of 50 μ M (~0.8 mg/ml) based on the SEC-MALS results. However, concentrations of 3.3-12 mg/ml HEPN^{Lpg} seem relatively high for bacteria. Therefore, the dimer-tetramer equilibrium of WT HEPN^{Lpg} determined by SEC-MALS remains controversial, and we removed the SEC-MALS data.

Alternatively, analytical ultracentrifugation (AUC) showed that HEPN^{Lpg} WT exists in dimer-tetramer equilibrium in solution. Moreover, surface plasmon resonance (SPR) experiments revealed that HEPN^{Lpg} WT exhibits considerably fast association and dissociation rates, indicating that the self-assembly process is transient and reversible. These results suggest that tetrameric HEPN^{Lpg} is always present in the HEPN^{Lpg} WT sample, thereby interfering with its RNase activity (as illustrated in the accompanying figure).

Figure 2

As per the reviewer's suggestion, we used dimeric HEPN^{Lpg} (Q64A) to conduct RNase assays with mutagenesis experiments. Specifically, all the mutants were redesigned based on this dimeric HEPN^{Lpg} variant, with the Q44 and E47 residues mutated to alanine. As a result, the dimeric HEPN^{Lpg} effectively digested 23S rRNA and 16S rRNA at lower concentrations (80–160 nM) (as illustrated in the accompanying figure).

Total RNA	+	+	+	+	+	+	+
HEPN (nM)	+	10	20	40	80	160	-
RNase A (U)	-	-	-	-	-	-	1

Additionally, the dimeric HEPN^{Lpg} Q44A, E47A, and Q44AE47A mutants showed significant decreases in RNase activity compared to dimeric HEPN^{Lpg}. However, the dimeric HEPN^{Lpg} R98A and H103A mutants had negligible impacts on RNase activity. Furthermore, the removal of metal ions almost completely abolished the RNase activity of dimeric HEPN^{Lpg}, with Mg²⁺ being the only metal ion that restored the RNase activity of EDTA-treated dimeric HEPN^{Lpg}. The catalytic properties of HEPN^{Lpg} and its variants were also re-evaluated using the dimeric form. The results indicated that the k_{cat}/K_m significantly decreased in the dimeric HEPN^{Lpg} mutants (Q44A, E47A, and Q44AE47A) and EDTA-treated dimeric HEPN^{Lpg} compared to the dimeric HEPN^{Lpg}. In conclusion, these findings suggest that the Mg²⁺-coordinated site is the active site of the HEPN^{Lpg} ribonuclease.

Accordingly, we revised and stated the results in the Results section titled “HEPN^{Lpg} possesses a new catalytic site” (third paragraph); Additionally, we replaced Figure 4a, b, Supplementary Figure 3 and Supplementary Table 2 with the new versions below.

Figure 4

Supplementary Figure 3

Supplementary Table 2. Relative RNase activity of dimeric HEPN^{Lpg} and its variants

HEPN ^{Lpg}	V_{max}	k_{cat}^a	K_m^a	k_{cat}/K_m
Dimeric HEPN^{Lpg}	27.18	271.80	0.69	393.91
Dimeric HEPN^{Lpg}+EDTA	0.93	9.34	0.80	11.68
Dimeric HEPN^{Lpg} Q44A	1.99	19.88	0.92	21.61
Dimeric HEPN^{Lpg} E47A	1.69	16.86	1.01	16.69
Dimeric HEPN^{Lpg} Q44AE47A	0.98	9.83	0.95	10.35

^aThe units for k_{cat} and K_m are defined as RFU min⁻¹ μM⁻¹ and μM, respectively.

3) To test the specificity of RNA cleavage, total RNA from *E. coli* could be used for comparison. Could HEPN(Lpg) also cleave mRNA as *S. oneidensis* HEPN (Yao et al, 2015) or tRNA as *A. flos-aquae* HEPN (Songailiene et al, 2020)? The resolution of the gel in Fig. 2B is not sufficient, if only a few nucleotides are cleaved from tRNA, as observed for *A. flos-aquae* HEPN. If the target is ribosomal RNA, can it be cleaved in ribosomes, as they exist in the cell?

To monitor the stoichiometry of HEPN(Lpg) and MNT(Lpg), native PAGE was performed mixing MNT(Lpg) with tetrameric HEPN(Lpg) or dimeric HEPN(Lpg): it is not clear which dimeric mutant was used. Native PAGE suggests that a fairly stable complex is formed between HEPN(Lpg) and MNT(Lpg), but the resolution of this method is limited. Therefore I propose to use SEC-MALS to characterize HEPN(Lpg) WT and its dimeric mutant (Q64A R73A) binding to MNT(Lpg) using different protein ratios. In vitro AMPylation experiment of HEPN(Lpg) dimeric mutant Q64A R73A could directly show that only the HEPN(Lpg) tetramer is AMPylated.

Answer: Thank you for insightful suggestions.

First, we performed an RNA cleavage assay using total RNA extracted from the *E. coli* B strain. Despite the inability to determine precise sequence similarity due to the lack of sequencing data for ribosomal RNA from this strain, HEPN^{Lpg} was still able to

effectively digest 23S rRNA and 16S rRNA, similar to its activity in *Legionella pneumophila* (data not shown). This result suggests that HEPN^{Lpg} may possess broad sequence specificity, where the recognition of structural features in RNA molecules is more critical than the recognition of specific nucleotide sequences. The detailed mechanistic action of HEPN^{Lpg} RNase needs further investigation through comprehensive structural studies.

Additionally, HEPN^{Lpg} was able to digest the corresponding 23S rRNA and 16S rRNA from the intact 70S ribosome purified from an *E. coli* B strain (purchased from New England Biolabs). In this case, some digested RNA remained as relatively large fragments, possibly because rRNA is tightly packed and protected by ribosomal proteins, making many regions inaccessible to RNases. Furthermore, to visualize small-sized rRNA (5S rRNA and tRNA), we used a 12% acrylamide gel, which indicated that HEPN^{Lpg} had no impact on 5S rRNA or tRNA across variable concentrations. The HEPN toxin from *S. oneidensis* was shown to digest specific *ompA* mRNAs (Yao et al., 2015); however, the detailed methods for identifying the specific mRNA targets remains unclear. Consequently, we did not investigate specific mRNA digestion in this study; this investigation will require further research.

Accordingly, we stated the results in the Results section titled “HEPN^{Lpg} possesses a new catalytic site” (lines 4–11 of p. 7) and the Discussion section titled “HEPN^{Lpg} acts as a Mg²⁺-dependent ribonuclease” (lines 24–25 of p. 13 and 1–2 and 17–18 of p. 14); and added new figures (Supplementary Figure 1).

Supplementary Figure 1

For native-PAGE, we used HEPN^{Lpg} Q64A to analyze dimeric HEPN^{Lpg}. Annotations have been added to the legends of Supplementary Figure 7c.

To further examine the stoichiometry of HEPN^{Lpg} and MNT^{Lpg}, we employed size exclusion chromatography (SEC) with several standard components. Due to the

resolution limitations of the gel-filtration column (Superdex 200 10/300 GL), distinguishing between the peaks of the tetrameric HEPN^{Lpg}-MNT^{Lpg} complex (76 kDa) and tetrameric HEPN^{Lpg} (64 kDa) was challenging. To address this issue, we identified complex formation based on the proportion of remaining MNT^{Lpg}. When the sample contained an excess of MNT^{Lpg} relative to HEPN^{Lpg} (e.g., HEPN^{Lpg}:MNT^{Lpg}=1:5), the excess MNT^{Lpg} could interfere with the interpretation of the results. Therefore, we incubated HEPN^{Lpg} (tetrameric or dimeric) with MNT^{Lpg} at a 1:1 ratio for 30 minutes before injection into the column. The results showed that MNT^{Lpg} binds to HEPN^{Lpg} exclusively in its tetrameric state in solution, not in its dimeric state. Moreover, unlike tetrameric HEPN^{Lpg}, dimeric HEPN^{Lpg} cannot be poly-AMPylyated by cognate MNT^{Lpg}. Collectively, these findings suggest that MNT^{Lpg} binds solely to tetrameric HEPN^{Lpg} to induce the poly-AMPylation reaction.

Accordingly, we stated the results in the Results section titled "HEPN^{Lpg} could be poly-AMPylyated by MNT^{Lpg} in vivo and in vitro" (lines 6–8 of p. 11) and "Crystal structural insight into the MNT^{Lpg}-HEPN^{Lpg} complex" (lines 12–15 of p. 13); and the Discussion section titled "MNT^{Lpg} modulates HEPN^{Lpg} despite lacking the α 4 helix" (lines 3–5 of p. 16); and added new figures (Figure 8d and Supplementary Figure 6a).

Figure 8

d

Supplementary Figure 6

Reviewer #3

Comments to the Author

This review of the manuscript by Jin and Jeon et al. focuses solely on the structural work presented. The structures appear to be of high quality and there are no apparent anomalies in the data collection or refinement statistics (Supp Table 3). It is also appreciated that a 2Fo-Fc omit map is shown for the AMPylated HEPN structure. Below are additional minor edits the authors should consider.

-it would be helpful to also report the buried surface area as a percentage of the entire surface area in addition to the value in square Angstroms

Answer: Thank you for your comment. As suggested, we have included the values corresponding to the surface area (\AA^2), buried surface area (\AA^2), and buried surface area percentage for the crystal structures of HEPN^{Lpg} WT, HEPN^{Lpg} Q64A, and the HEPN^{Lpg}-MNT^{Lpg} complex in Supplementary Table 1.

Supplementary Table 1. Surface area of the crystal structures of the HEPN^{Lpg}-MNT^{Lpg} module

Type of interface	HEPN ^{Lpg} WT	Surface area (Å ²)		Entire surface area (Å ²)	Buried surface Area (Å ²)	Buried surface area percentage (%)
Dimeric	Chains A/B	Chain A: 7214	Chain B: 7339	14553	882	6.1
	Chains C/D	Chain C: 7121	Chain D: 7265	14386	874	6.1
Tetrameric	Chains A/D	Chain A: 7214	Chain D: 7265	14479	405	2.8
	Chains B/C	Chain B: 7339	Chain C: 7121	14460	419	2.9
	Chains A/C	Chain A: 7214	Chain C: 7121	14335	311	2.2
Type of interface	HEPN ^{Lpg} Q64A	Surface area (Å ²)		Entire surface area (Å ²)	Buried surface Area (Å ²)	Buried surface area percentage (%)
Dimeric	Chains A/B	Chain A: 7490	Chain B: 7607	15097	928	6.1
Type of interface	HEPN ^{Lpg} -MNT ^{Lpg} complex	Surface area (Å ²)		Entire surface area (Å ²)	Buried surface Area (Å ²)	Buried surface area percentage (%)
Dimeric	Chains A/B	Chain A: 7744	Chain B: 7167	14911	848	5.7
	Chains C/D	Chain C: 7438	Chain D: 6941	14379	975	6.8
Tetrameric	Chains A/D	Chain A: 7744	Chain D: 6941	14685	430	2.9
	Chains B/C	Chain B: 7167	Chain C: 7438	14605	434	3.0
	Chains A/C	Chain A: 7744	Chain C: 7438	15182	273	1.8
HEPN-MNT complex	Chians A/E	Chain A: 7744	Chain E: 6224	13968	512	3.7
	Chains C/E	Chain C: 7438	Chain E: 6224	13662	629	4.6

-All structure figures: Cyan and pale cyan lettering is very hard to read on a computer

Answer: We appreciate the reviewer's suggestion. We have revised all the cyan and pale cyan letters in the panels of Figures 1a-c, 3d-e, 7a-b, 8a-b and Supplementary Figure 7a-b, 8a, d, and 9a-c for improved presentation.

-Supplementary figure 2 (and Discussion): Can the authors state whether or not there are any crystal contacts or other structural features that may influence the angle between helices a2 of the cleft?

Answer: Thank you for bringing up this point. No crystal contacts with symmetric mates were observed in the crystal structures of either tetrameric or dimeric HEPN^{Lpg}. Additionally, the angles between the $\alpha 2$ helices were nearly identical in both states of HEPN^{Lpg}. This result suggests that the angle between the $\alpha 2$ helices is primarily determined by the interaction between the two counterpart $\alpha 2$ helices. Furthermore, tetramerization did not significantly influence the angle between the $\alpha 2$ helices in HEPN^{Lpg} (as illustrated in the accompanying figure).

Accordingly, we stated the results in the Discussion section titled “Molecular basis of oligomeric state transition in HEPN^{Lpg}” (lines 24–25 of p. 14 and lines 1–3 of p. 15).

Point-by-point response to the referee's comments

Reviewer(s)' Comments to Author:

Reviewer #1:

Comments to the Author

The authors have addressed most of my concerns. I have one question as follows: Fig3b, In vitro total RNA digestion assay. The length in each band in the marker should be labelled. Can the total RNA completely cleaved by RNase A? No cleavage products can be seen in the gel.

Answer: Thank you for pointing this out. As suggested by the reviewer, we labeled the bands in the marker in Fig. 3b. Furthermore, RNase A demonstrated high efficiency in total RNA cleavage assays, producing non-specific cleavage patterns. Consequently, total RNA was degraded into very small fragments or individual nucleotides. After extensive digestion, these small RNA fragments migrated rapidly and were not retained on the gel matrix, or they were not visible as distinct bands. This likely accounts for the absence of detectable cleavage products in the lane of RNase A, as illustrated in Fig. 3b.

Reviewer #2:

Comments to the Author

The additional data included after the revision support the structural part of the manuscript. However, I am still concerned about the introduction of the new type active site in the well characterized HEPN fold. This is an important statement and since RNases are common contaminants in protein preparations, the RNase activity of the proposed new type active site should be double checked.

1. New data on the RNA cleavage by the dimeric versions of the Q44A, E47A, Q44AE47A mutants support their importance for the observed RNA cleavage activity. However, it is not clear from the methods whether or not identical expression/purification protocols were used for the Q64A and these mutants of the new type active site. It is now stated that:

p.19: „The recombinant HEPNLpg WT protein and its mutants were overexpressed in E. coli 13 Rosetta2 (DE3) cells using Luria–Bertani (LB) medium supplemented with kanamycin. E. coli Rosetta2 (DE3) cells harboring HEPNLpg WT and its

mutants (except for Q64A and R73A) were grown until the OD₆₀₀ reached 0.5–0.6, after which the proteins were induced with 0.5 mM isopropyl β-D-1-thiogalactopyranoside (IPTG) for 4 h at 37°C. Due to the high toxicity, the cells harboring HEPNLpg Q64A and R73A were grown to the stationary phase (OD₆₀₀ of ~ 1–1.2) and induced by 0.5 mM IPTG for only 2 h.“

If, as stated, the Q64A and Q64A+Q44A, Q64A+E47A, Q64A+Q44AE47A mutants were expressed under different conditions, possible contamination in the Q64A and R73A protein preparations with RNases (detected by the RNase Alert Kit used to compare RNase activity) cannot be excluded.

Answer: Thank you for pointing this out. First, we apologize for any misunderstanding caused by the insufficient explanation in the “Methods” section of our manuscript. For over-expression of target protein, *E. coli* Rosetta2 (DE3) cells harboring HEPN^{Lpg} WT were grown until the OD₆₀₀ reached 0.5–0.6, after which the proteins were induced with 0.5 mM isopropyl β-D-1-thiogalactopyranoside (IPTG) for 4 h at 37°C. In our revised work, several HEPN^{Lpg} mutants were redesigned on the basis of the dimeric HEPN^{Lpg} (Q64A) to identify the active site of HEPN^{Lpg}. In this regard, the cells harboring dimeric HEPN^{Lpg} (Q64A and R73A) were grown to the stationary phase (OD₆₀₀ of ~1–1.2) and induced by 0.5 mM IPTG for only 2 h. Meanwhile, to ensure consistent induction conditions, all HEPN^{Lpg} mutants potentially related to the active site (Q64AR98A, Q64AH103A, Q64AQ44A, Q64AE47A, and Q64AQ44AE47A) were induced in the same manner as HEPN^{Lpg} Q64A. Furthermore, the entire process of purification was identical for all HEPN^{Lpg} samples, including both HEPN^{Lpg} WT and its mutants.

Accordingly, we revised the method in the Methods section titled “Protein expression and purification” (lines 24–25 of p. 19 and line 1 of p. 20).

Furthermore, regarding the potential contamination of the Q64A and R73A protein preparations with RNases, we compared the RNase activities of HEPN^{Lpg} Q64A and R73A with that of RNase A under consistent assay conditions. Given that the RNaseAlert™ kit is carefully optimized to detect several RNase targets, including RNase A, we used RNase A as a positive control for this comparison. As a result, the RNA substrates were rapidly digested by RNase A (almost completely digested at the beginning of the assay), whereas HEPN^{Lpg} Q64A and R73A moderately cleaved the RNA substrates over time (as illustrated in the accompanying figure). This indicates that the RNase activities detected for HEPN^{Lpg} Q64A and R73A were due to inherent properties of those samples, not influenced by other RNases. Therefore, we can exclude the possibility of contamination by other RNases.

2. Substrate specificity. In the final schema in Fig. 9 the authors claim that HEPNLpg cleaves only ribosomal 16S and 23S rRNAs. However, HEPNLpg also cleaves synthetic RNA substrate used for the mutant activity assay and they did not test mRNA cleavage, so this is not correct.

Methods, In vitro RNase assay, p.25 RNase Alert Kit, not rRNA, was used for mutant activity characterization: „To conduct a quantitative analysis of the RNase activity of apo HEPNLpg, a fluorescence quenching assay was conducted on dimeric HEPNLpg and its variants using an RNase Alert Kit (IDT, Coralville, IA, USA)“. This kit is designed to test to test RNase contaminations and the supplier states that „The sequence of the RNaseAlert™ Substrate has been carefully optimized to detect several RNases, including RNase A, RNase T1, RNase I, micrococcal nuclease, S1 nuclease, mung bean nuclease, and Benzonase™“ (from <https://www.thermofisher.com/order/catalog/product/AM1964>).

The cleavage of this substrate by the Q64A mutant indicates that HEPN is not specific for ribosomal RNA and can also cleave other RNA substrates. Thus, the activity observed in Fig. 3b and Fig. S3 is a non-specific RNase activity (no specific cleavage pattern is observed and all rRNA is degraded). However, the authors did not test mRNA cleavage, so this remains to be elucidated.

Therefore „23S rRNA & 16S rRNA cleavage“ on the right side of the schema (Fig. 9) should be changed to „RNA cleavage“.

Answer: We agree with the reviewer's point. We used an RNase Alert Kit to quantitatively compare the RNase activity of HEPN^{Lp9} and its mutants by measuring RFU (endpoints) and catalytic properties (kinetics). However, as the reviewer mentioned, this kit does not utilize specific rRNA sequences or tertiary structures but rather a universal sequence that can be digested by several RNases. Additionally, our experimental results showed that while HEPN^{Lp9} does cleave 23S rRNA and 16S

rRNA, it exhibits non-specific cleavage patterns. Therefore, from this study, we concluded that HEPN^{Lpg} cleaves RNA in a non-specific manner.

Furthermore, previous research has demonstrated that HEPN from *S. oneidensis* specifically cleaved *ompA* mRNA. However, in this study, we were unable to determine whether HEPN^{Lpg} can cleave any specific mRNA. Thus, based on the *in vitro* RNase assay results, it is difficult to rule out the possibility that HEPN^{Lpg} cleaves mRNA in a non-specific manner. This investigation will require further research. Therefore, to avoid misunderstanding, we revised the content of Figure 9 to represent “RNA cleavage” as suggested by the reviewer.

Accordingly, we stated the result in the Discussion section titled “HEPN^{Lpg} acts as a Mg²⁺-dependent ribonuclease” (lines 6–10 of p. 14).

3. Testing of the *in vivo* toxicity of both canonical and new active site mutants (Q64A, Q64A+R98A, Q64A+H103A, Q64A+Q44A, Q64A+E47A) is required as an alternative method and would definitely support new type active site hypothesis. Since Q64A is toxic, mutations in the correct active site should abolish its toxicity. This can be done by spot-plating assay comparing the growth of serial dilutions of the *E. coli* cells expressing mutants in the presence and absence of IPTG (+1% glucose in the absence of IPTG to prevent background expression from T7 promoter) (see in doi.org/10.7554/eLife.98528.1).

Answer: Thank you for the valuable suggestions. In accordance with the reviewer's recommendation, we conducted spot-plating assays using the dimeric HEPN^{Lpg} (Q64A) and its mutants. The results showed that the induction of HEPN^{Lpg} Q64A, Q64AR98A, and Q64AH103A in *E. coli* cells significantly arrested cell growth on plates. In contrast, cell growth was rescued when HEPN^{Lpg} Q64AQ44A, Q64AE47A, and Q64AQ44AE47A were expressed *in vivo*. These results strongly support the hypothesis of a new type of active site, which consists of a Mg²⁺-coordinated site.

Accordingly, we stated the result in the Results section titled “HEPN^{Lpg} possesses a new catalytic site” (lines 4–7 of p. 9) and added the method in the Methods section titled “Spot-plating assays” (lines 7–16 of p. 26); and added new figures (Supplementary Figure 4).

Supplementary Figure 4

Minor comments:

1. p.17: Not all mutants are listed in the methods, double mutants (like Q44A+Q64A, E47A+Q64A, etc) are not:

„The mutations in HEPN^{L-pg} were R98A, H103A, Q44A, E47A, Q44AE47D, Q64A and R73A, while the mutations in MNTL^{L-pg} were G36AS37T, D48E, D50E, and D48ED50E.“

Q44AE47D mutant should be Q44AE47A.

Answer: Thank you for pointing this out. Since all the HEPN^{L-pg} mutants potentially related to the active site were redesigned on the basis of dimeric HEPN^{L-pg} (Q64A), we corrected the mutants to Q64AR98A, Q64AH103A, Q64AQ44A, Q64AE47A, and Q64AQ44AE47A in lanes 12–13 of p. 19.

2. Fig. S3: no units for the enzyme activity (y axis) and substrate concentration (x axis) are shown.

Answer: Thank you for pointing this out. We added the units for enzyme activity and substrate concentration to Fig. S3.

Reviewer #3:

The authors sufficiently addressed the initial review comments. The revised manuscript is acceptable in its current form.

We acknowledge the concerns raised by Reviewer #2 regarding potential RNase contamination, as this is a common issue encountered in laboratory settings. However, based on our experimental methods and results, we have outlined the following points to address and mitigate these concerns.

-The RNase A used in our *in vitro* total RNA digestion assay and *in vitro* RNase assay was not prepared in-house, but was purchased from Roche (Prod. No.: 10109142001). The purity of the final sample used in the *in vitro* RNase assays was evaluated via SDS-PAGE, as shown in the accompanying figure.

If the RNase activity observed in the RNase assay were due to contamination by RNases such as RNase A, the dimeric HEPN^{L-pg} sample and the RNase A sample used as a positive control should exhibit the same RNA digestion pattern in the *in vitro* total RNA digestion assay (Figure 3b). However, the results demonstrate that the dimeric HEPN^{L-pg} sample selectively cleaves 23S rRNA and 16S rRNA in a non-specific manner, whereas the RNase A sample digests all components of total RNA, including 23S rRNA, 16S rRNA, 5S rRNA, and tRNA. Furthermore, as shown in Figure 4b, the RNase activity of dimeric HEPN^{L-pg} significantly decreases following EDTA treatment, but is restored with the addition of Mg²⁺ ions. In line with the reasoning mentioned above, if the RNase activity in the HEPN^{L-pg} sample were due to contamination by an RNase such as RNase A, which is metal-independent, it would not exhibit sensitivity to EDTA treatment or Mg²⁺-dependent changes in RNase activity.

-To assess the potential impact of contamination from other RNases in the *in vitro* RNase assays, we employed the Ribolock RNase inhibitor (henceforth, Ribolock) (Thermo Scientific™, Cat. No.: EO0382), which effectively inhibits the activity of various RNases. 1 U Ribolock was added to HEPN^{L-pg} samples (including dimeric HEPN^{L-pg} and its mutants), and we compared the RFU values between two groups: Ribolock-treated and non-treated samples. The results showed almost no variation in RFU between the two groups, indicating that the RNase activities observed in the

HEPN^{Lpg} samples were unlikely to result from contamination by other RNases (as illustrated in the accompanying figure).

-The results of the *in vivo* spot-plating assays support the findings from the *in vitro* RNase assays regarding the active site of HEPN^{Lpg}. Furthermore, by identifying the active site of HEPN^{Lpg} through assessing the impact of its expression on cell growth *in vivo*, we can eliminate the possibility of contamination by other RNases that may have occurred during protein purification or the assay process.

-All pipette tips and tubes used in the *in vitro* RNase assay and *in vitro* total RNA digestion assay were autoclaved, and the laboratory bench surfaces and pipettes were treated with RNaseZap™ RNase Decontamination Solution (Invitrogen, Cat. No.: AM9780) to prevent RNase contamination.

- We have included several published papers where the RNase Alert Kit (IDT) was used to validate the RNase activity of target proteins. This includes two publications from our group and one from another research group.

1> In-Gyun Lee et al., Structural and functional studies of the *Mycobacterium tuberculosis* VapBC30 toxin-antitoxin system: implications for the design of novel antimicrobial peptides, *Nucleic Acids Research*, 2015 Sep 3;43(15):7624-37. doi: 10.1093/nar/gkv689.

2> Do-Hee Kim et al., Role of Pemi in the *Staphylococcus aureus* PemIK toxin antitoxin complex: Pemi controls PemK by acting as a PemK loop mimic, *Nucleic Acids Research*, 2022 Feb 28;50(4):2319-2333. doi: 10.1093/nar/gkab1288.

3> Basem Al-Shayeb et al., Diverse virus-encoded CRISPR-Cas systems include streamlined genome editors, *Cell*, 2022 Nov 23;185(24):4574-4586.e16. doi: 10.1016/j.cell.2022.10.020.

Overall, considering the factors outlined above, it is unlikely that the RNase activity observed in the HEPN^{Lpg} sample is attributable to RNase contamination.